# GAF-dependent chromatin plasticity determines promoter usage to mediate locust gregarious behavior

Xiao Li [1,2,3,6], Feng Jiang [1,2,6], Qing Liu[1], Zizheng Zhang[1,2], Wenjing Fang[1,2], Yutong Wang [1], Hongran Liu[1,2] & Le Kang [1,2,3,4,5]✉

## Abstract

**Locusts, as devastating pests, can reversibly transform between solitary individuals and gregarious swarms with markedly different behaviors. Epigenetic regulation orchestrated by changes in chromatin openness modulates behavioral plasticity by controlling gene expression. However, the gene regulation mechanisms by which chromatin openness controls behavioral changes remain largely unknown. Here, we explored the regulatory function of chromatin openness in modulating behavioral plasticity, in which the remodeler GAF regulated brain-specific promoter usage in locusts. The increased chromatin openness in gregarious locusts initiated transcription of the brain-specific promoter of *henna*, a critical gene in dopamine synthesis and gregarious behavior mediation. Furthermore, GAF-dependent chromatin openness responded coordinately to population density changes. Fragment mutagenesis abolished *henna* promoter activity due to the dysfunction of the GAF-binding site. Mechanistically, the three GAF-binding sites played a synergetic role in remodeling chromatin openness and activating transcription initiation. Our study reveals a novel epigenetic mechanism linking chromatin regulation with behavioral polyphenism in insects during environmental changes.**

**Keywords** Open Chromatin; Promoter Usage; Transcription Initiation; Behavioral Polyphenism; Locusts
**Subject Category** Evolution & Ecology

## Introduction

Phenotypic plasticity, the capacity of a single genome to produce distinct phenotypes under different environmental conditions, exhibits clear epigenetic regulatory characteristics (Yoon et al, 2023). The epigenetic mechanism is the major regulator of gene expression in organisms, thereby playing a crucial role in the formation of alternative phenotypes (Wang and Kang, 2014). Polyphenism, one of the most extreme forms of phenotypic plasticity in insects, serves as a crucial survival strategy, enabling organisms to adapt to dynamic environments throughout their lifespan (Kang et al, 2004; Yoon et al, 2023). The migratory locust, *Locusta migratoria*, a worldwide important pest species, displays remarkable polyphenism by exhibiting reversible phases response to changes of population density (Wang and Kang, 2014). Phase changes from solitary to gregarious locusts lead to the formation of large-scale swarms and devastating plague outbreaks. Despite having identical genotypes, the solitary and gregarious locusts exhibit striking phenotypic differences in morphology, skeleton, coloration, behavior, physiology, reproduction, and immunology (Guo et al, 2020b; He et al, 2016; Ma et al, 2011; Wang and Kang, 2014; Wang et al, 2013). In particular, behavioral changes occur within hours when the locusts aggregate together (Burrows et al, 2011). Therefore, the phase change of locusts in behavior exhibits the phenomenon of short-term epigenetic regulation.

In the past two decades, many studies focus on post-transcriptional regulation in locust phase changes (Guo et al, 2011, 2015, 2020a; Ma et al, 2011; Yang et al, 2014). Compared to their solitary locusts, gregarious locusts exhibit elevated concentrations of dopamine and increased gene expression of *henna*, *pale*, and *vat1* in the brain (Ma et al, 2011). Especially, the post-transcriptional regulation of *henna* plays a critical role in locust behavioral changes, because *henna* encodes an important enzyme, phenylalanine hydroxylase (PAH), in dopamine synthesis (Guo et al, 2015). MicroRNA-133 exerts subtle regulation of *henna* expression by targeting its coding sequence to promote gregarious behavior (Yang et al, 2014). A long noncoding RNA, *PAHAL*, interacts with SRSF2 at the 5'UTR of nascent *henna* RNA and *Gypsy* retrotransposon to promote *henna* expression (Zhang et al, 2020, 2022). The pre-transcriptional regulation in modulating henna expression associated with behavioral phase changes remains unknown, although its importance in initiating gene expression and post-transcriptional regulation has been demonstrated (Furlan et al, 2021).

Chromatin dynamics linked with chromatin openness are critical for pre-transcriptional regulation because they directly

[1]State Key Laboratory of Integrated Management of Pest Insects and Rodents, Institute of Zoology, Chinese Academy of Sciences, Beijing, China. [2]University of Chinese Academy of Sciences, Beijing, China. [3]Guangzhou National Laboratory, Guangzhou, China. [4]College of Life Science, Hebei University, Baoding, China. [5]Institute of Cell and Gene Technology, Shenzhen University of Advanced Technology, Shenzhen, China. [6]These authors contributed equally: Xiao Li, Feng Jiang. ✉E-mail: lkang@ioz.ac.cn

impact the recruitment of transcription factors and the transcriptional machinery in nucleosome-depleted regions (Balsalobre and Drouin, 2022). Open chromatin regulation is involved in determining the phenotypic plasticity in several insect species, because of its ability to create a chromatin environment conducive to promoter selection and transcription initiation (Muller and Tora, 2014). Thousands of caste-specific differences in open chromatin regions, the vast majority of which are located within introns, have been identified among queens, workers, and drones in adult honey bees (Lowe et al, 2022). Moreover, chromatin openness is strongly correlated to the biased gene expression between queen and worker larvae of honey bees (Zhang et al, 2023). These studies focus on the genome-wide comparison of open chromatin sites and the downstream genes whose expression is controlled by open chromatin regulation. However, the chromatin remodeler responsible for regulating transcription initiation, which in turn drives the development of different plastic phenotypes, remains elusive.

In this study, we explored the regulatory role of chromatin remodeler in epigenetic mechanisms associated with short-term chromatin regulation and behavioral plasticity in locusts. First, we investigated the critical open chromatin region to control the transcription initiation of *henna*. Then, we proved its regulatory activity of chromatin openness by using cell line experiments in vitro and CRISPR/Cas9-mediated fragment mutagenesis in vivo. Moreover, we identified the *henna* promoter, which is predominantly involved in behavioral modulation, to explore phase-related transcription initiation in gregarious locusts. Finally, we determined the chromatin remodeler involved in modulating phase-related chromatin openness and explored its regulatory role in chromatin openness mechanistically. Our study reveals open chromatin dynamics in regulating behavioral changes of locusts and provides insights into the epigenetic mechanisms underlying regulation of behavioral polyphenism in insects.

# Results

## Differentiated chromatin openness determines promoters of *henna*

Because of the critical role of *henna* in behavioral plasticity (Ma et al, 2011), we explore the open chromatin landscape and identified the open chromatin regions that are in the genomic neighborhood of *henna*. Firstly, we generated a genomic map of DNase I hypersensitive sites (DHSs) from the brains of gregarious locusts using DNase-Seq sequencing. Approximately 95% of the sequencing reads were mapped to the locust genome in a proper pair manner. The metagene plots of DHS signals showed an enrichment upstream of protein-coding genes and a good concordance of the oligo-capping captured transcription start sites (TSSs), indicating a high-quality representation of *cis*-regulatory regions identified by DNase-Seq sequencing (Fig. 1A). The peak calling analysis identified a total of genome-wide 34,633 DHS peaks. The *henna* locus contains five DHS peaks, all of which are located within the gene body region of *henna*. Importantly, there were two DHS peaks, DHS1 and DHS2, were co-localized with exons of *henna* (Fig. 1B).

To determine the predominant regulatory region among these five DHS peaks, we profiled the histone modification distribution of

three critical chromatin marks that collectively reflect epigenetic characteristics of active promoters (Fig. 1B). Except for DHS0, which is associated with the long noncoding RNA PAHAL (Zhang et al, 2020), DHS1 exhibited the highest signals of H3K4me3 and H3K27ac among the identified DHS peaks. Moreover, the deposition of H3K36me3 occurred downstream of DHS1 and spread into the coding region of *henna*. We next quantified the chromatin openness of DHS1 and DHS2 using DNase I-qPCR assays. Compared to the inaccessible control, DHS1 ($P = 0.0025$) but not DHS2 ($P = 0.9667$), exhibited a significantly higher DHS intensity in the brains of gregarious locusts (Fig. 1C). The dual-luciferase reporter assays showed that DHS1 exhibited significantly higher promoter activities than DHS2 in vitro (Fig. 1D, $P < 0.0001$). To prove the in vivo promoter role of DHS1 in gregarious locusts, we disturbed the DHS1 function by using CRISPR/Cas9-mediated fragment mutagenesis (Figs. 1E and EV1A). As expected, we observed a significant reduction in both mRNA ($P < 0.0001$) and protein levels ($P = 0.0003$) of *henna* in the brains of homozygous DHS1-dysfunctional mutants of gregarious locusts, compared to wild-type controls. In contrast, the mRNA ($P = 0.5308$) and protein ($P = 0.5895$) levels of *henna* did not show significant changes in the fat bodies of homozygous DHS1-dysfunctional mutants of gregarious locusts. Moreover, to assess the influence of DHS1 on transcription initiation per se, we probed the expression of nascent RNAs by quantifying the chromatin-associated RNAs, which are freshly synthesized RNAs and remain bound to transcription sites. The results showed that, compared to wild-type controls, the nascent RNA expression in DHS1 was significantly reduced in the brains of the homozygous DHS1-dysfunctional mutants of gregarious locusts, while its did not change significantly in DHS2 (Fig. EV1B, for TSS1, $P = 0.0286$, Mann–Whitney test; for TSS2, $P = 0.4221$). The intensity of chromatin openness of DHS1 in the brains of gregarious locusts was significantly higher than that in solitary locusts (Fig. 1F, $P = 0.0497$), whereas this difference was not observed for DHS2 ($P = 0.8128$). Therefore, the differentiated chromatin openness of DHS1 determined the utilization of promoter usage that initiated the phase-related transcription of *henna* in brains.

## Phase-related transcriptional initiation of *henna* promoter responds to changes in population density

To explore the pre-transcriptional regulation of *henna* via chromatin regulation, we examined transcription initiation using 5'-Cap captured RNA-seq (Jiang et al, 2019; Liu et al, 2021). Three TSSs, corresponding to three core promoters, initiated the transcription of three distinct isoforms. These isoforms share a large majority of exons but have different mRNA leaders and partial protein sequences at the N-terminal tails (Fig. 2A). To determine promoter usage of *henna*, we quantified the expression levels of the three TSSs in different tissues and organs by 5'-Cap RNA-seq (Fig. 2B). The TSS1 and TSS2 exhibited predominant transcription in the brains (Transcripts Per Kilobase Million, TPM = 121.53) and the fat bodies (TPM = 166.91), respectively. The TSS3 showed restricted expression only in gonads with limited transcription activity (TPM = 31.70). The TSS1 and TSS2 expressions determined by 5'-Cap RNA-seq and the qRT-PCR assays were confirmed in both brains and fat bodies of gregarious and solitary locusts (Figs. 2C and EV2). Both TSS1 and TSS2 contained

an initiator element in the dominated point of the TSS Clusters (Appendix Fig. S1). The two isoforms governed by the TSS1 and TSS2 showed a 16 amino acid difference at the N-terminal tail, thereby resulting in differences in a phosphorylated site, serine-9, and three-dimensional structures (Appendix Figs. S1 and S2). Given PAH protein catalyzes L-Phe in a tetrahydrobiopterin ($BH_4$)-dependent hydroxylation (Flydal et al, 2019), we calculated the Gibbs free energy for binding cofactors $BH_4$ and L-Phe in these two isoforms. L-Phe and BH4 preferred to occupy the active sites of the TSS1 and TSS2 isoforms, respectively (Fig. 2D,E). Thus, the alternative promoter selection of *henna* may result in structurally and functionally distinct enzymes in different tissues.

To investigate the roles of promoter usage of *henna* in response to the phase change of locusts, we then examined the RNA expressions of TSS1 and TSS2 in solitary and gregarious locusts by TSS-specific RT-qPCR. The TSS1 expression was significantly higher in gregarious locusts than in solitary locusts in the brains, but not in the fat bodies (Fig. 2F,G, $P = 0.0453$ and $P = 0.9698$, Student $t$ tests). The TSS2 expression was significantly lower in the brains of gregarious locusts than in those of solitary locusts, whereas its expression was not significantly changed in the fat bodies between the two-phase locusts (Fig. 2F,G, $P = 0.0414$ and $P = 0.9190$, Student $t$ tests). Since 4 h of isolation or crowding is sufficient to induce changes in the mRNA level of *henna* (Ma et al, 2011; Yang et al, 2014), we set the 4-h time point after isolation or crowding to quantify the pre-transcriptional expression change of *henna*. The TSS1 expression in the brains exhibited a significant decrease when isolating gregarious locusts ($P = 0.0408$, Student's $t$ test) and a significant increase when crowding solitary locusts ($P = 0.0116$, Student's $t$ test). The changes of the TSS1 expression in the fat bodies were similar to those observed in the brains (Fig. 2G, $P = 0.0004$ for G vs IG, $P = 0.0033$ for S vs CS, Student's $t$ test). In contrast, the changes of population density did not induce the correspondent expression changes of TSS2 in the brains and fat bodies (Fig. 2F, $P = 0.9991$ and 0.2878 for brains; Fig. 2G, $P = 0.7009$ and 0.8701 for fat bodies, Student's $t$ test). Thus, phase-related transcription initiation of the *henna* promoter in brains sensitively responded to the changes of locust population density.

## Dominant transcription initiation of *henna* promoter promotes aggregative behavior

To determine the effects of transcription initiation on behavioral traits, the TSS-specific knockdown of TSS1 and TSS2 was performed using double-stranded RNA interference (RNAi, Fig. EV3A). The RT-qPCR analysis showed high efficiency and high specificity of expression silencing using TSS-specific knockdown (Fig. EV3B). The TSS1-specific knockdown resulted a significant transition from gregarious behavior toward solitary behavior in behavioral arena assays (Fig. 3A, $P < 0.0001$). Specifically, the TSS1-specific knockdown in gregarious locusts led to a significant reduction in distance moved (Fig. 3B, $P = 0.0003$) and a marginal significant reduction attraction index ($P = 0.0411$). In contrast, there is no significant difference in the probabilistic metric of gregariousness index (Pgreg), distance moved, and attraction index in the gregarious locusts subjected to TSS2-specific knockdown (for distance moved, $P = 0.6405$, for attraction index, $P = 0.0813$). To investigate the TSS1-specific

expression contributions to the total expression abundance of *henna*, we examined the mRNA and protein levels of *henna* in brains. The mRNA ($P = 0.0286$) and protein ($P = 0.0323$) levels were reduced by more than 80% and 50% after 48 h of TSS1-specific knockdown (Fig. 3C,D). However, the TSS2-specific expression only slightly altered the total expression abundance of *henna* at mRNA ($P = 0.6890$) and protein ($P = 0.7411$) levels. Thus, the transcription initiation of TSS1 in brains contributed dominantly to the *henna* expression and to the formation of gregarious behavior.

## GAF activates the promoter transcription to mediate gregarious behavior

To determine the transcription factor of TSS1, the transcription factor binding sites (TFBS) were predicted to explore the regulator that invoke the regulatory function of DHS1. A total of 126 TFBSs, a large majority of which were the binding sites of transcriptional activators, were identified (Appendix Fig. S3A). Due to the unevenness of TFBS distribution, the DHS1 sequence was divided into three subregions based on the K-means clustering of TFBS density (Fig. 4A). The promoter activities of the three subregions were then quantified in vitro using dual-luciferase reporter assays. Deletion of any of the three subregions resulted in a significant decrease of promoter activities ($Ps < 0.0001$, Brown–Forsythe and Welch ANOVA tests), indicating a critical role for transcriptional initiation of these subregions (Fig. 4B). Moreover, appearance of DHS1 subregions II resulted in a significant increase of promoter activity, while the other two didn't ($Ps < 0.0001$, Brown–Forsythe and Welch ANOVA tests). To determine the crucial DHS1 subregion for regulating chromatin openness, the intensity of chromatin openness was quantified by DNase-qPCR (Fig. 4C). Of the three subregions, DHS1 sub-region II exhibited the highest intensity of chromatin openness ($P = 0.0012$, Student's $t$ test). Therefore, the DHS1 sub-region II as a core component is crucial for the promoter integrity of *henna* in brains.

The greater expression variation of TSS1 implied that the transcriptional regulation of TSS1 was more sophisticated than that of housekeeping genes, *actin5c* and *gapdh* (Appendix Fig. S3B). Assuming that increased sophistication in transcriptional regulation requires more specific regulators, we attempted to identify such regulators in DHS1 sub-region II by excluding shared TFBSs between DHS1 and the promoter regions of *actin5c* and *gapdh*. A total of three TFBSs, corresponding to six transcription factors (TFs), were found to be specific for DHS1 (Fig. 4D). Intriguingly, multiple binding sites of GAGA factor (GAF, LOCMI11683) and Grainy head (Grh, LOCMI12694) were located within DHS1 sub-region II (Fig. 4E). To determine the function of GAF and Grh in governing transcription initiation of TSS1, we quantified their mRNA expression in multiple tissues and organs (Fig. 4F). GAF and Grh was predominantly expressed in the brain and in the antennae, respectively ($Ps < 0.0001$, Student's $t$ test). Moreover, the TSS1 expression had a positive correlation solely with the GAF expression, while no such correlation was observed with the Grh expression in tissues (Appendix Fig. S4). Notably, no GAF-binding site was identified in TSS2, whereas the Grh binding sites were predicted in both TSS1 and TSS2, indicating a non-exclusive role of Grh (Fig. 4G). GAF in the locust has a central C2H2-type zinc finger domain (Appendix Fig. S5), which is responsible for DNA

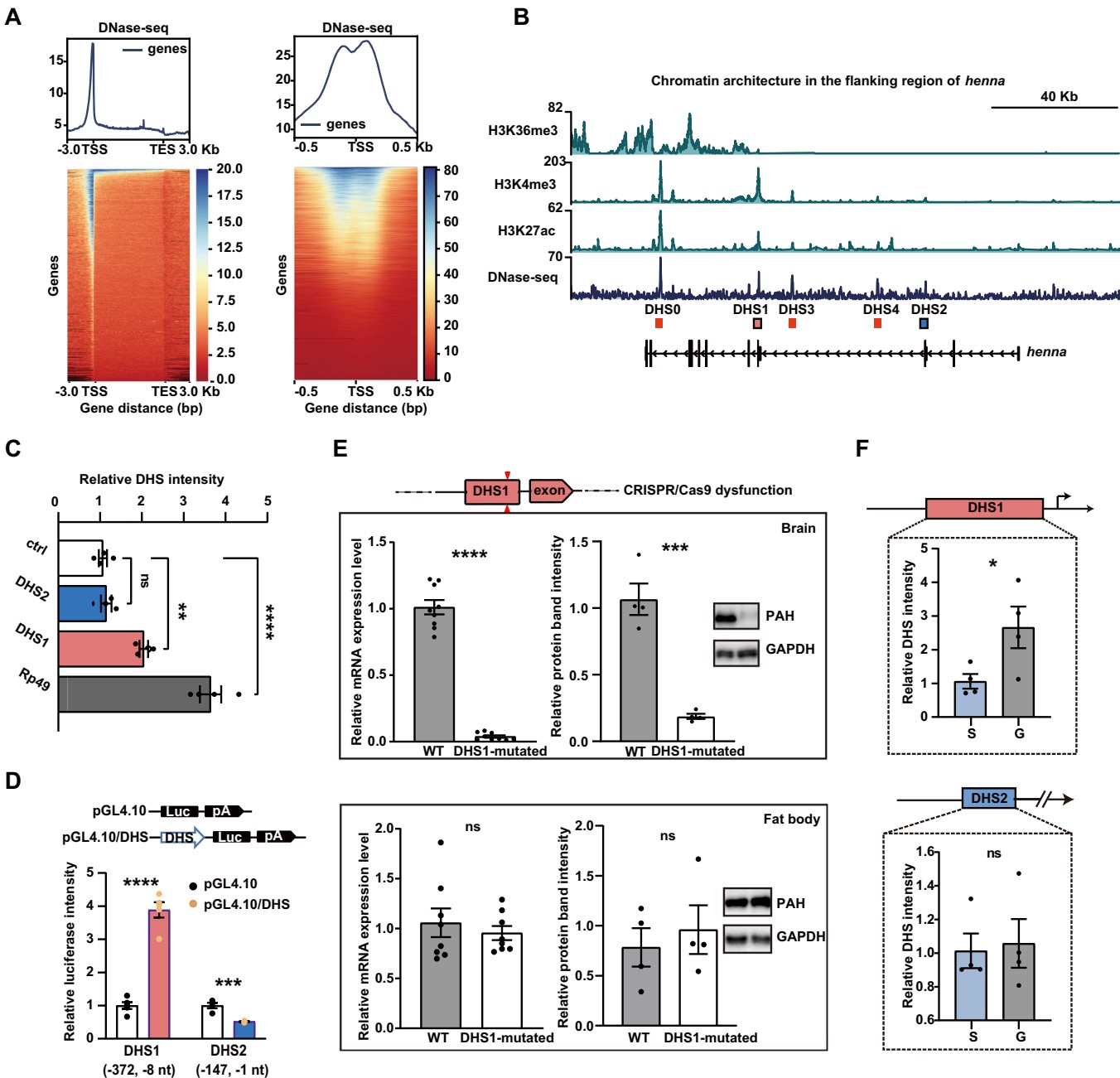

**Figure 1. Differentiated chromatin openness determines the alternative promoter usage of *henna* in brains.**

(A) Metagene profiles and heatmaps of DNase I hypersensitive sites. Left: The transcription start sites (TSSs) and transcription end sites (TESs) were retrieved from the official gene sets of the locust genomes. Right: The oligo-capping captured TSSs are used as the reference points. (B) The chromatin environment flanking *henna* gene in brains. H3K36me3 represents transcription elongation at the gene body, while H3K4me3 and H3K27ac denote active transcription markers. Five regions of DNase-seq peaks represent the open chromatin regions. DHS1 and DHS2 co-localized with *henna* exons. Histone modification data were retrieved with an accompanying paper. Scale bar, 40 kb. (C) The relative intensity of chromatin openness for DHS1 and DHS2 ($n = 4$ for each group). An inaccessible chromatin region is for a negative control. The DHS of *rp49* is for a positive control. P values from left to right, $P = 0.9667$, $P = 0.0025$, $P = 7.89e$-5. (D) The promoter activity for DHS1 and DHS2 in *S2* cells ($n = 5$ for DHS1, and 4 for DHS2). P values from left to right, $P = 3.22e$-6, $P = 0.0014$. (E) mRNA and protein levels of PAH for wild type (WT) and homozygotes with DHS1-mutated in the brains and fat bodies (At protein levels, $n = 4$ for each group. At mRNA levels, $n = 8$ for each group in fat bodies, and 9 in brains). In the brains, P values from left to right, $P = 5.55e$-12, $P = 0.0003$. In the fat bodies, P values from left to right, $P = 0.5308$, $P = 0.5895$. (F) Chromatin openness at DHS1 and DHS2 in solitary and gregarious locusts ($n = 4$ for each group). P values from top to bottom, $P = 0.0497$, $P = 0.8128$. Values are means and error bars indicate SEM. *$P < 0.05$, **$P < 0.01$, ***$P < 0.001$, ****$P < 0.0001$ (C–F, Student's *t* tests). Source data are available online for this figure.

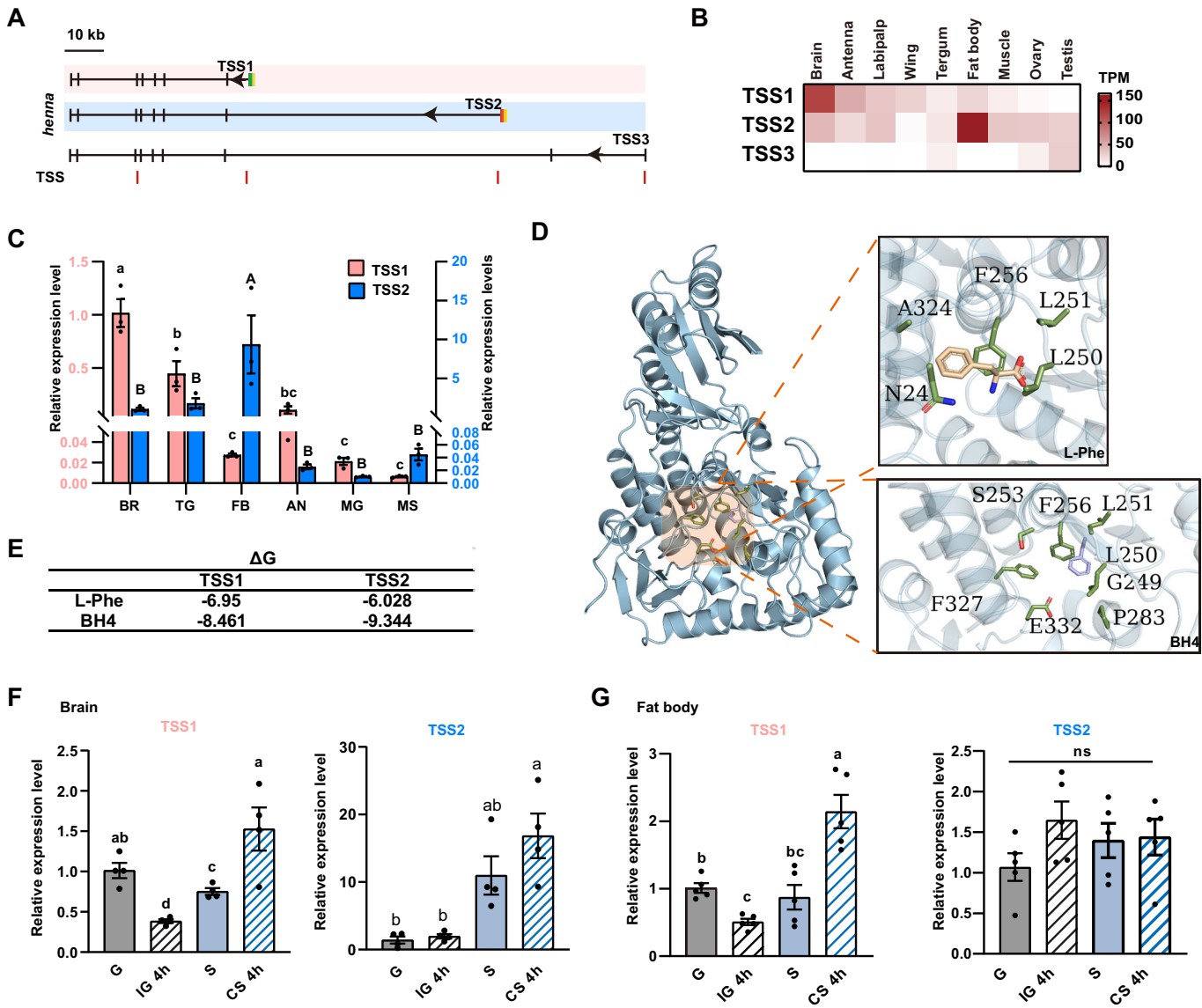

**Figure 2. Phase-related transcriptional initiation of alternative *henna* promoter responds to the changes of population density of locusts.**

(A) The gene structure of henna in brains of gregarious locusts. Four transcription start sites (TSSs) are localized within the *henna* gene body. Among these TSSs, three of them are co-localized with the 5′ ends of *henna* isoforms. Scale bar, 10 kb. (B) Expression of three *henna* TSSs in locusts. The expression level is measured using transcripts per million reads (TPM). (C) Expression of *henna* TSS1 and TSS2 determined by TSS-specific qPCR in gregarious locusts ($n = 3$ for each group). BR brain, TG thoracic ganglia, FB fat body, AN antenna, MG midgut, MS muscle. (D) Detailed view of the active site of *henna* in complex with L_Phe (wheat) and BH4 (light blue). Relevant residues are labeled. (E) Gibbs free energy ($\Delta G$) of binding cofactors, $BH_4$ and L-Phe, of *henna* TSSs. (F) Abundance of transcriptional initiation of TSSs in the brain of gregarious (G), solitary (S), isolated gregarious (IG 4 h), and crowded solitary (CS 4 h) locusts ($n = 4$ for each group). (G) Abundance of transcriptional initiation of *henna* TSSs in the fat body of gregarious (G), solitary (S), isolated gregarious (IG 4 h), and crowded solitary (CS 4 h) locusts ($n = 5$ for each group). Values are means and error bars indicate SEM. *$P < 0.05$, **$P < 0.01$, ***$P < 0.001$, ****$P < 0.0001$. The same letters above the bars indicate no significant difference between groups. Different letters indicate a significant difference between groups. Combined letters indicate groups are different from others but not from each other. (C, F) Ordinary one-way or (G) Brown–Forsythe and Welch ANOVA tests. Source data are available online for this figure.

binding to the GAGAG pentanucleotide (Chetverina, Erokhin and Schedl, 2021). In the brain, the DNA binding capacity of locust GAF remained in concordance with its nucleus localization, substantiated by the assays of immunofluorescence staining and subcellular western blotting (Appendix Fig. S6). In vitro, GAF showed a direct interaction with DHS1 via the predicted GAF-binding sites, as proven by the electrophoretic mobility shift assay (EMSA) (Fig. 4H). Moreover, the chromatin immunoprecipitation

PCR (ChIP-PCR) assays showed that the GAF binding in the homozygous DHS1-dysfunctional mutants was significantly more impaired than that in the wild-type controls (for WT, $P = 0.0018$; for DHS1-mutated, $P = 0.8726$, Figs. 4I and EV4). Therefore, GAF possesses potential function in regulating transcription initiation of TSS1 in brains.

We verified the function relationships between GAF and TSS1 by the examination of TSS1 expression after the knockdown of

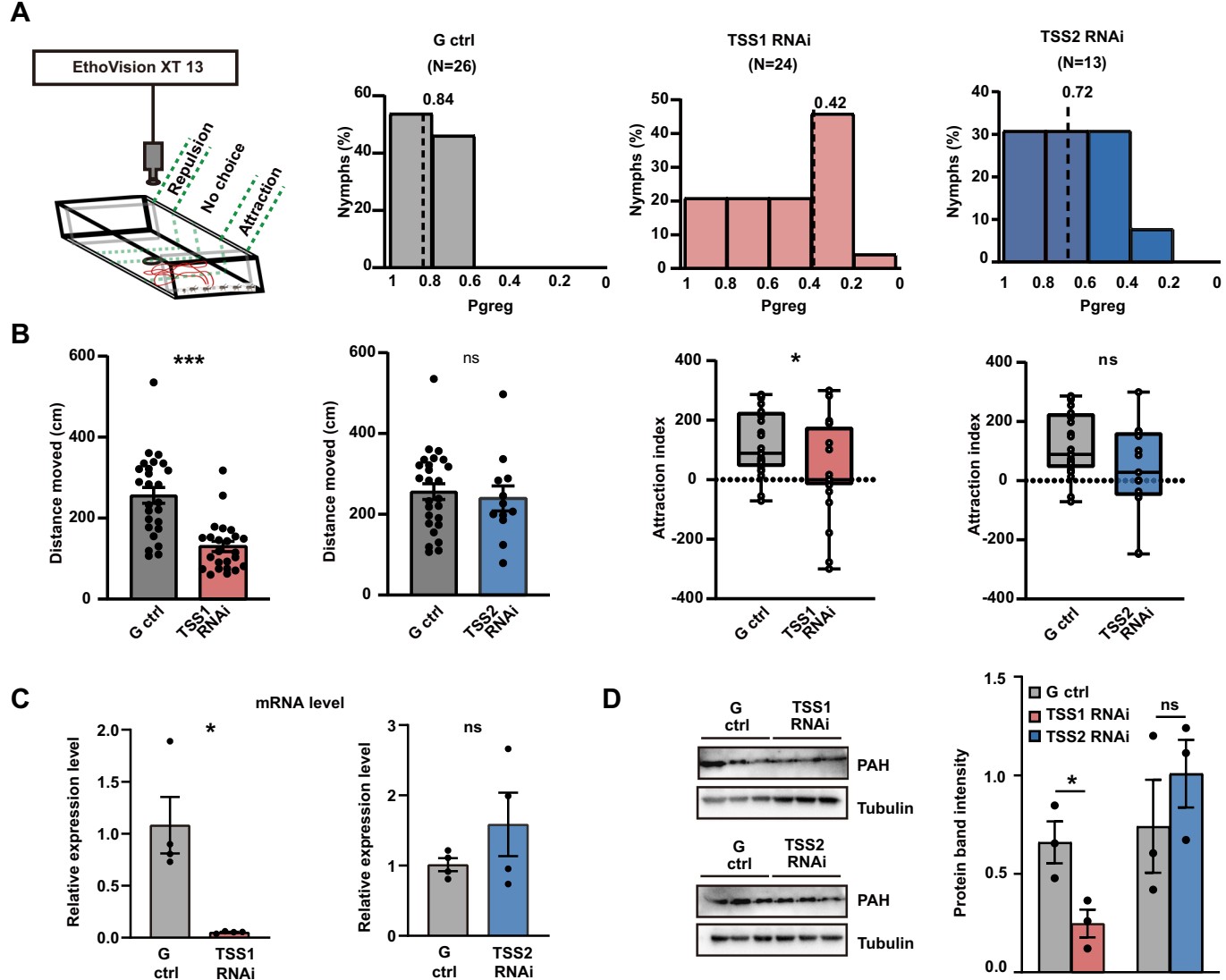

**Figure 3. The dominant transcriptional initiation of *henna* in brains maintains the gregarious behaviors.**

(A) A schematic diagram of a behavioral arena assay (left). Thirty gregarious 4th nymphs are set as stimuli. G ctrl indicates gregarious phase states. $N = 26$ for G ctrl, $n = 24$ for TSS1 RNAi, and $n = 13$ for TSS2 RNAi. $P < 0.0001$ for TSS1 RNAi, $P = 0.7823$ for TSS2 RNAi, Wilcoxon rank sum exact tests. Dashed lines represent the medians of Pgreg. (B) The distance moved and attraction index for G ctrl, TSS1, and TSS2 RNAi. $N = 26$ for G ctrl, $n = 24$ for TSS1 RNAi, and $n = 13$ for TSS2 RNAi. $P$ values from left to right, $P = 0.0003$, $P = 0.6405$, $P = 0.0411$, $P = 0.0813$. The attraction indexes are plotted using a box-whisker plot: the whiskers represent the maximum and minimum values, the boxes represent the upper quartile, median, and lower quartile, and the dots represent data points. (C) The mRNA levels of G ctrl, TSS1 RNAi and TSS2 RNAi ($n = 4$ for each group). $P$ values from left to right, $P = 0.0286$, $P = 0.6890$. (D) The protein levels of G ctrl, TSS1 RNAi and TSS2 RNAi ($n = 3$ for each group). $P$ values from left to right, $P = 0.0323$, $P = 0.7411$. Values are means and error bars indicate SEM. $*P < 0.05$, $***P < 0.001$. (B–D, Student's $t$ tests, and C, left, Mann–Whitney test). Source data are available online for this figure.

GAF expression. The knockdown of GAF expression in gregarious locusts significantly decreased the TSS1 expression by at least 70% (Fig. 4J), accompanied by a reduction of Pgreg, distance moved, and attraction index (Fig. 4K,L, for distance moved, $P = 0.0472$, and for attraction index, $P = 0.0262$). As anticipated, the TSS2 expression was not affected in gregarious locusts subjected to GAF knockdown ($P > 0.0500$, Student's $t$ test). To exclude the possibility that GAF interacts to other genes in the dopamine synthesis pathway, we predicted GAF-binding sites in the upstream regions of *pale*, *Ddc*, and *tan*. Despite identifying a limited number

of GAF-binding sites (Appendix Fig. S7), their location outside open chromatin regions inhibited their regulatory potential of GAF with these genes in vivo, thereby indicating an exclusive role of GAF specifically on TSS1. Meanwhile, our genomic analyses of several holometabolous and hemimetabolous insects, which diverged ~300 million years ago, revealed the widespread presence of multiple GAF-binding sites in the flanking region of henna TSSs (Appendix Fig. S8). To ascertain whether GAF is co-localized with *henna* within specific brain regions, we performed dual immuno-fluorescence staining by co-labeling of GAF and PAH. Both GAF

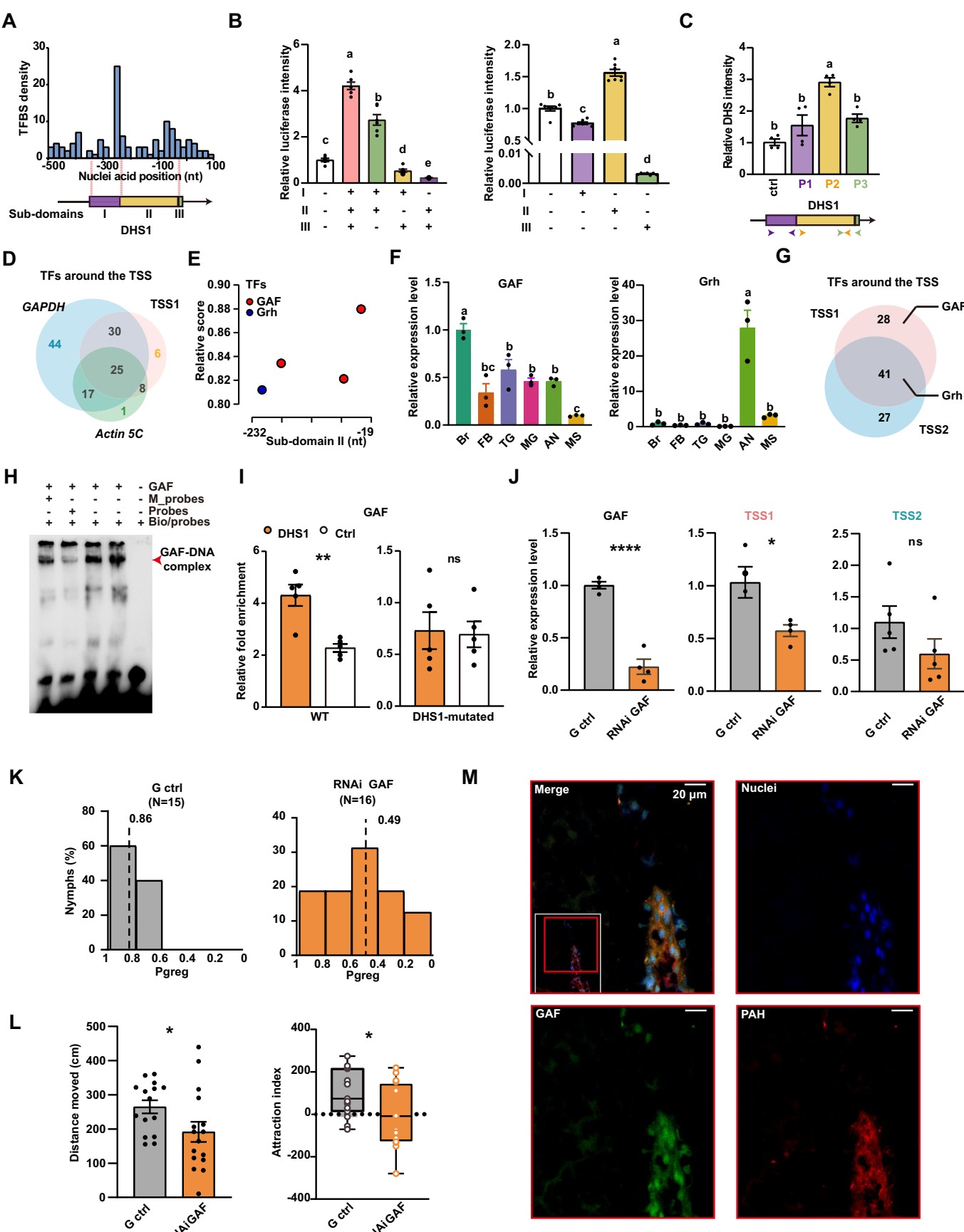

**Figure 4. GAGA factors activate the phase-related expression of _henna_ TSS1 at the promoter.**

(A) The distribution of TFBSs at DHS1. Based on the density of TFBSs, the DHS1 was divided into three subregions I (purple), II (yellow), and III (green) by K-means clustering. (B) The promoter activities of DHS1 subregions in S2 cells ($n = 6$ for each group in the left, and 7 for its in the right). $Ps < 0.0001$, Brown–Forsythe and Welch ANOVA tests. (C) The relative DHS intensity in vivo at subregions ($n = 4$ for each group). Colored arrows represent the locations of primers. The primer for region III includes a fragment of region II. (D) The Vann diagram of TFs around TSSs of _TSS1_, _GAPDH_, and _actin5c_. (E) The location of three GAF and one Grh binding sites at region II of DHS1. (F) The expression pattern for GAF and Grh in tissue or organs of gregarious locusts ($n = 3$ for each group). GAF is present only at TSS1, while Grh is found at both TSSs. (G) The Vann diagram of TFs at _henna_ TSS1 and TSS2. (H) The direct interaction between GAF and its binding sites on DHS1 analyzed using EMSA. Bio/probes are biotin-labeled DHS1, while M_probes are DHS1 with mutated GAF-binding sites. The red arrow indicates the target band. (I) The fold enrichment of GAF binding at DHS1 in vivo in gregarious and DHS1-mutated nymphs proved by chromatin immunoprecipitation PCR ($n = 5$ for each group). Ctrl represents an inaccessible chromatin region. $P$ values from left to right, $P = 0.0018$, $P = 0.8726$. (J) mRNA expressions for GAF, TSS1, and TSS2 after the GAF expression reduction ($n = 4$ for GAF and TSS1, and 5 for TSS2). $P$ values from left to right, $P = 6.63e-5$, $P = 0.0269$, $P = 0.1870$. (K) Behavioral arena assays for G ctrl and RNAi GAF. $N = 15$ for G ctrl, and $n = 16$ for RNAi GAF. $P = 0.0004$, Wilcoxon rank sum exact test. Dashed lines represent the medians of _Pgreg_. (L) The distance moved and attraction index for G ctrl and RNAi GAF. $N = 15$ for G ctrl, and $n = 16$ for RNAi GAF. $P$ values from left to right, $P = 0.0472$, $P = 0.0262$. The attraction index are plotted with a box-whisker plot: the whiskers represent the maximum and minimum values, the boxes represent the upper quartile, median, and lower quartile, and the dots represent data points. (M) Immunofluorescence of PAH and GAF proteins in the brains of 4th-instar nymphs of gregarious locusts. Blue, Hoechst; Green, GAF; Red, PAH. Scale bar, 20 µm. Values are means and error bars indicate SEM. *$P < 0.05$, **$P < 0.01$, ****$P < 0.0001$ (C, F, ordinary one-way ANOVA tests; H–J, L, Student's t tests). The same letters above the bars indicate no significant difference between groups. Different letters indicate a significant difference between groups. Combined letters indicate groups are different from others but not from each other. Source data are available online for this figure.

and PAH proteins are localized in the neuronal cell body of the protocerebrum (Fig. 4M), which is involved in transforming sensory information into behavioral outputs (Phillips-Portillo and Strausfeld, 2012). Therefore, GAF specifically activated TSS1 expression in brains by directly binding to DHS1 of _henna_ to mediate gregarious behavior.

## GAF60 site is the core binding site for GAF-dependent regulatory activity

To investigate the special function of GAF-binding sites in regulating TSS1 transcription, the plasmids containing either one or all of the mutated GAF-binding sites (GAF28, GAF60 and GAF166) in the DHS1 sub-region II were constructed for assaying promoter activities (Fig. 5A). The mutation of GAF60 resulted in the greatest reduction in promoter activity ($Ps < 0.0001$, Student's t test). Accordingly, introducing any one of a single GAF-binding site into DHS2 also enhanced its promoter activity ($Ps < 0.0001$, Student's t tests), and the highest activity was observed in GAF60. Thus, the three GAF-binding sites in the DHS1 sub-region II displayed a synergistic effect in boosting promoter activities.

To verify that the promoter activities of these binding sits are regulated by GAF-dependent activation, the pGL4.10/DHS1 reporter plasmids were co-transfected with the pAc5.1/V5-His vector containing GAF from locusts (Fig. 5B,C). The heterologous expression of GAF significantly promoted the promoter activity of pGL4.10/DHS1 (Fig. 5C, $P = 0.0008$). In contrast, the activator activity of GAF was abolished when all GAF-binding sites were mutated (Fig. 5C, $P = 0.4254$). To further explore the activating effects of each GAF-binding site, the reporter plasmids fused with a single mutated GAF-binding site were co-transfected with overexpressed GAF. When GAF was overexpressed, the promoter activity of DHS1 increased significantly (Fig. 5D, $P < 0.0001$). Furthermore, among of the three GAF-binding sites, only the promoter activity of GAF60 mutants was significantly deprived ($P < 0.0001$), indicating that GAF60 plays the most crucial role in the activation of TSS1. The promoter activity of _henna_ DHS2 without any GAF-binding sites was not affected by GAF overexpression ($P = 0.1149$). Moreover, the insertion of any of the three GAF-binding sites into DHS2 significantly increased the promoter activities of DHS2 upon over-expressed GAF (Fig. 5E, for GAF28, $P = 0.0001$, for GAF60, $P = 0.0003$, and for GAF166, $P < 0.0001$). The increased levels were comparable

among the three GAF-binding sites (1.6-fold for GAF28, 1.4-fold for GAF60, and 1.6-fold for GAF166), indicating that the crucial role of GAF60 in DHS1 is dependent on the flanking sequence context. Therefore, in the context of multiple GAF-binding sites within DHS1, GAF60 was the most crucial binding site for the regulatory activity of the promoter mediated by GAF.

## Population density changes impact nucleosome remodeling of DHS1 via GAF28 and GAF166 binding

Because DHS1 showed greater chromatin openness in the brains of gregarious locusts than in solitary locusts, we explored the role of GAF in modulating chromatin openness using a parallel assay of ChIP-qPCR (chromatin immunoprecipitation quantitative polymerase chain reaction) and DNase-qPCR (Fig. 6A). The significant increases of histone H3 signal, representing nucleosome occupancy, were observed in the three DHS1 subregions after the GAF knockdown (Fig. 6B, $Ps < 0.0100$). The quantity of GAF binding at the _henna_ TSS1 decreased after 4 h of isolation of gregarious locusts (Fig. 6C, $P = 0.0042$) and increased after 4 h of crowding of solitary locusts (Fig. 6C, $P = 0.0292$). In addition, the intensities of chromatin openness were significantly reduced in DHS1 sub-region II containing GAF28 and GAF60, and in DHS1 sub-region III near the TSS1 (Fig. 6D, for GAF28 and GAF60, $P = 0.0084$; for sub-region III, $P = 0.0278$). To verify that the locust GAF can remodel nucleosome organization, we purified the GAF protein through recombinant expression and then incubated it with pre-assembled chromatin of DHS1 in vitro (Fig. 6E). The gradual depletion of di-nucleosomes compared to mono-nucleosomes with the increases of GAF amount indicated that GAF reduced the density of positioned nucleosomes of DHS1 (Fig. 6F; Appendix Fig. S9, for 0 vs 1, $P = 0.0214$; for 0.1 vs 1, $P = 0.0432$; for 0.5 vs 1, $P = 0.0012$). To determine the role of each GAF-binding site in nucleosome remodeling, we assembled the chromatin of DHS1 with mutated GAF-binding sites (Fig. 6G). GAF incubation resulted in the increase of DNA amounts in the nucleosome-free region (NFR) for the wild-type DHS1 (Lane 1 in Fig. 6G), while it did not affect the nucleosome array for DHS1 with all three GAF-binding sites mutated (Lane 2 in Fig. 6G). Compared to DHS1 with all three GAF-binding sites mutated, the restoration of both GAF28 and GAF166 showed increased DNA amounts in the nucleosome-free

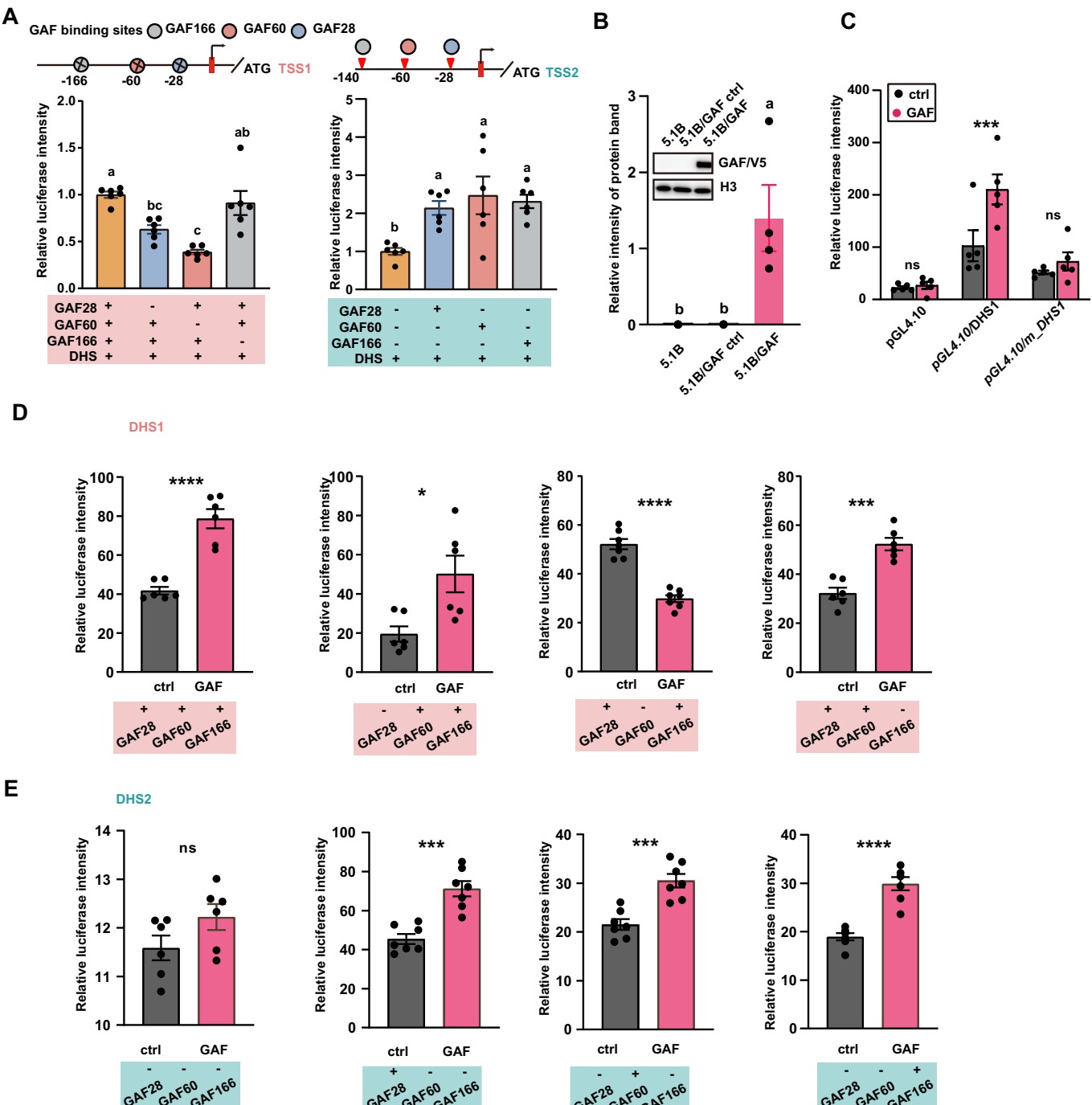

region (Lane 4 in Fig. 6G). However, the presence of either GAF28 or GAF166 alone with GAF60 in DHS1 failed to rescue this effect (Lanes 3 and 5 in Fig. 6G). Therefore, these results demonstrated a combined role of GAF28 and GAF166 in GAF-dependent nucleosome remodeling of DHS1.

To confirm whether nucleosome remodeling of DHS1 subregions responds to the changes in the population density of locusts, we first compared their chromatin openness between gregarious and solitary locusts in brains. The intensities of chromatin openness within DHS1 in gregarious locusts were significantly higher than those in solitary locusts

(Fig. 6H, $P = 0.0005$ for sub-region I; $P = 0.0229$ for GAF166; $P = 0.0136$ for GAF28 and GAF60; $P = 0.0209$ for sub-region III). There is no significant difference in chromatin openness of DHS2 between gregarious and solitary locusts, being in line with the absence of GAF-binding sites in DHS2 (Fig. 1F, $P = 0.8128$). We further isolated gregarious locusts and crowded solitary locusts, respectively, to quantify the differences in chromatin openness in response to changes of population density. The intensities of chromatin openness decreased significantly at GAF28 and GAF60 in DHS1 sub-region II and at DHS1 sub-region III (Fig. 6I, $P = 0.0468$ for GAF28 and GAF60; $P = 0.0017$ for

**Figure 5. GAF60 binding site is dominant to the promoter activation.**

(A) The promoter activities in *S2* cell for mutated *henna* DHS1 and inserted DHS2. The schematics of the constructed sequences are depicted in the upper column charts ($n = 6$ for each group). The mutation of GAF-binding sites replaced "GAG" with "CTC". For TSS2, the insertion of GAF-binding site is located at the same position as it in TSS1, except for the GAF166 at −140 bp. (B) The protein levels of expressional plasmids. The reverse complemental sequence of GAF is inserted for 5.1B/GAF ctrl as the negative reference ($n = 4$ for each group). (C) The change of transcriptional initiation of pGL4.10, DHS1, and DHS1 with mutated GAF-binding sites after the overexpression of GAF ($n = 5$ for each group). *P* values from left to right, $P = 0.5842$, $P = 0.0008$, $P = 0.4255$. (D) The transcriptional initiation levels of DHS1, GAF28⁻, GAF60⁻, and GAF166⁻ after the overexpression of GAF ($n = 6$ for DHS1, GAF28⁻, and GAF166⁻. $n = 7$ for GAF60⁻). *P* values from left to right, $P = 3.96e-5$, $P = 0.0128$, $P = 1.34e-6$, $P = 0.0002$. (E) The transcriptional initiation levels of DHS2, GAF28⁺, GAF60⁺, and GAF166⁺ after the overexpression of GAF ($n = 6$ for DHS2 and GAF166⁺; $n = 7$ for GAF28⁺ and GAF60⁺). *P* values from left to right, $P = 0.1149$, $P = 0.0001$, $P = 0.0003$, $P = 1.21e-5$. Values are means and error bars indicate SEM. *$P < 0.05$, **$P < 0.01$, ****$P < 0.0001$ (A, B, ordinary one-way ANOVA tests; C–E, Student's *t* tests). Same letters above the bars indicate no significant difference between groups. Different letters indicate a significant difference between groups. Combined letters indicate groups are different from others but not from each other. Source data are available online for this figure.

sub-region III) after 4 h of isolation of gregarious locusts. In contrast, the intensities of chromatin openness increased significantly at GAF28 and 60 in DHS1 sub-region II (Fig. 6J, $P = 0.0155$) and at DHS1 sub-region III (Fig. 6J, $P = 0.0215$) after 4 h of crowding of solitary locusts. Therefore, GAF-dependent nucleosome remodeling specially enhances the chromatin openness of DHS1 through the binding of GAF28 and GAF166 in response to changes in locust population density.

## Discussion

We present the first pre-transcriptional mechanism of locust behavioral plasticity by deciphering the landscape of open chromatin and performing sophisticated experiments. We demonstrate that the gregarious behavior of locusts is mediated by GAF-dependent regulation of chromatin openness on the brain-specific transcription initiation of the *henna* promoter through the synergistic collaboration of three GAF-binding sites. This study reveals a population density-dependent role of the chromatin remodeler in regulating the open chromatin of the *henna* promoter. The selection of brain-specific promoter usage confirms the importance of GAF in modulating the phenotypic plasticity of insects. Mechanistically, the three GAF-binding sites collaborate in the transcription initiation regulation of *henna* within 4 h, demonstrating the capacity for rapid GAF-dependent chromatin regulation of behavioral polyphenism in short timescales. This precise pre-transcriptional regulatory mechanism, derived from interactions between chromatin remodelers and transcription initiation, enlightens us to develop novel approaches for behavioral manipulation in insects.

Selection of *henna* promoter usage determines the organs/tissues where PAH enzymes function. We observed brain- and the fat body-specific transcription initiation of *henna* promoters in locusts. This alternative promoter selection of *henna* results in structurally distinct PAH enzymes in brain and fat body. The TSS1-driven *henna* isoform in brains, which has a unique serine-9 at the N-terminal tail, shares a large majority (96.5% of the total amino acids) of protein sequence with that driven by TSS2 in fat bodies. As a result, activation of the serine-16 phosphorylated enzyme by phenylalanine occurs more rapidly and at lower concentrations than that of the unphosphorylated enzyme (Kowlessur et al, 1995). Thus, the balance of active and inactive forms of PAH enzymes is modulated by serine-16 phosphorylation. Protein sequence comparison shows that serine-16, the primary phosphorylation site for the human PAH, closely aligns with serine-9 of the TSS1-driven *henna* isoform in locusts (Coleman and Neckameyer, 2004). PAH is the first hydroxylase enzyme in dopamine biosynthesis, and its deficiencies affect dopamine production and dopamine signaling pathways in brain (Holmqvist et al, 2012). Depending on the requirement for dopamine dynamics in the brain, the transition between active and inactive forms of PAH can occur rapidly within a few hours during behavioral changes in locusts. Accordingly, constitutive TSS2-driven *henna* expression is in accordance with the constant requirement for dopamine synthesis in fat body. Thus, population density changes of locusts only affect the transcription initiation of TSS1 in the brain but not that of TSS2 in fat body. This is consistent with that only TSS1 but not TSS2 is recruited to synthesize dopamine and mediate gregarious behaviors in brain of locusts. Taken together, pre-transcriptional selection of *henna* promoter usage produces the two PAH enzymes that are structurally and functionally distinct, providing an explanation for why GAF-dependent chromatin regulation of TSS1 but not TSS2 is exclusively engaged in brain of locusts.

The chromatin remodeler in the central nervous system governs phenotypic plasticity of insects (Simola et al, 2016). Our results reveal that the chromatin remodeler GAF enhances chromatin openness for transcription initiation in central nervous system during phase change of locusts. This process enables a reversal of the chromatin state through the action of chromatin remodelers in locusts, demonstrating the chromatin of *cis*-regulatory region is high dynamics and closely linked to pre-transcription regulation. Therefore, upon immediate sensory perception of aggregation pheromone (Guo et al, 2020b; Yang et al, 2023), behavioral responses in locusts depend on proper recruitment of chromatin regulators in the nervous system. In fact, processes of remodeling chromatin structure that regulate gene transcription are conserved epigenetic mechanisms by which the central nervous system accomplishes phenotypic plasticity in response to environmental inputs in insects. In the carpenter ant, the effects of environmental clues on caste specification are regulated by differential recruitment of the chromatin regulator CBP to chromatin in brains (Simola et al, 2013). CBP also plays an important role in caste development in worker and queen larval heads of honey bees (Wojciechowski et al, 2018). Specifically, the recruitment of CBP to chromatin depends on GAF deployment. GAF-binding sites significantly overlap with CBP binding sites, and knockdown of GAF expression reduces the association of RNA polymerase II with CBP (Boija et al, 2017; Holmqvist et al, 2012). Regardless of GAF or CBP, no matter which one plays a more crucial role, our study, along with these previous studies, underscores the importance of chromatin remodelers in the central nervous system in governing the phenotypic plasticity of insects in response to sensory input from external environmental cues. Despite the well-recognized role of CBP in the phenotypic plasticity of insects, we propose that GAF in

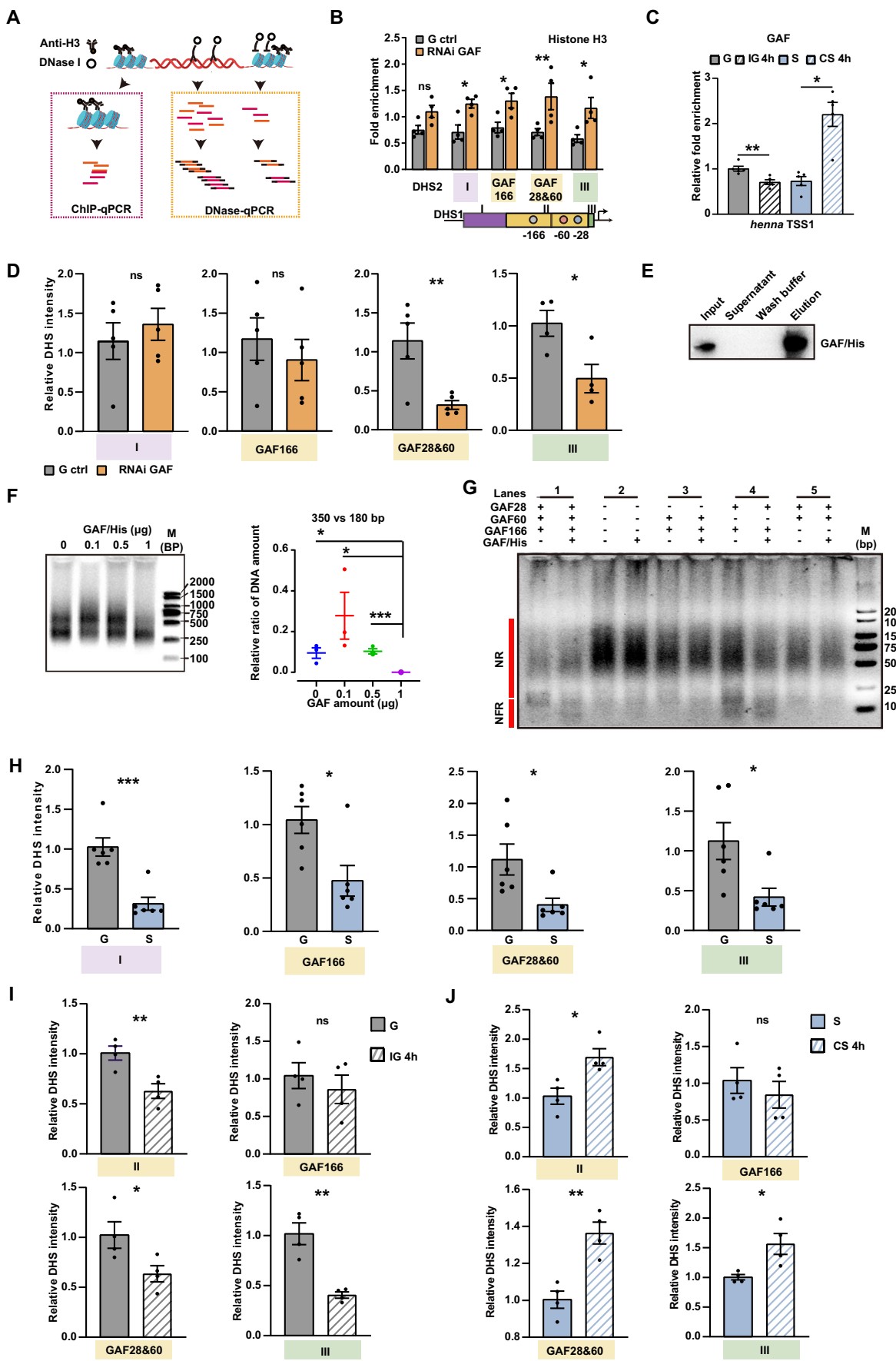

**Figure 6.  GAF contributes to the sustentation of the chromatin openness of DHS1 via GAF28 and GAF166.**

(A) A schematic diagram of ChIP-qPCR and DNase-qPCR to detect the intensity of chromatin openness. Histone H3 represents the intensity of nucleosome occupancy. (B) The chromatin-nucleosome-bound factions of DHS1 subregions and DHS2 in G ctrl and RNAi GAF ($n = 4$ for each group). P values from left to right, $P = 0.0816$, $P = 0.0271$, $P = 0.0271$, $P = 0.0073$, $P = 0.0205$ (Multiple unpaired t tests). (C) The quantities of GAF binding at *henna* TSS1 in brains of gregarious (G), isolated gregarious (IG 4 h), solitary (S), and crowding solitary (CS 4 h) nymphs ($n = 5$ for each group). P values from left to right, $P = 0.0042$, $P = 0.0292$. (D) The chromatin-openness intensities of DHS1 subregions in G ctrl and RNAi GAF ($n = 5$ for I, GAF166, and GAF28&60; $n = 4$ for III). P values from left to right, $P = 0.5091$, $P = 0.5001$, $P = 0.0084$, $P = 0.0278$. (E) Western blotting of the purified GAF/His protein. Anti-His-tag was used as a primary antibody. (F) An agarose gel electrophoresis showing the gradual depletion of tri-nucleosomes at DHS1 as the GAF/His amount increases. The relative ratio of DNA amounts for di-nucleosomes versus mono-nucleosomes (left) at DHS1 ($n = 3$ for each group) was determined using the Agilent 2100 Bioanalyzer system. P values from left to right, $P = 0.0214$, $P = 0.0432$, $P = 0.0012$. (G) An agarose gel electrophoresis of nucleosome-binding plasmids after GAF incubation and the digestion of MNase. Lanes 1 are the plasmid pGL4.10/DHS1. Lanes 2 are the plasmid pGL4.10/DHS1 with mutated GAF sites. Lanes 3 are the plasmid pGL4.10/DHS1 with mutated GAF28. Lanes 4 are the plasmid pGL4.10/DHS1 with mutated GAF60. Lanes 5 are the plasmid pGL4.10/DHS1 with mutated GAF166. NR, nucleosome region. NFR, the nucleosome-free region. (H) The chromatin openness of three DHS1 subregions in the brains of G and S locusts ($n = 6$ for each group). P values from left to right, $P = 0.0005$, $P = 0.0229$, $P = 0.0136$, $P = 0.0209$. (I) The chromatin openness of DHS1 subregions and GAF-binding sites in brains of G and IG 4 h locusts ($n = 4$ for each group). P values: $P(II) = 0.0095$, $P(GAF166) = 0.5038$, $P(GAF28\&60) = 0.0468$, $P(III) = 0.0017$. (J) The chromatin openness of DHS1 subregions and GAF-binding sites in the brain of S and CS 4 h locusts ($n = 4$ for each group). P values: $P(II) = 0.0155$, $P(GAF166) = 0.4690$, $P(GAF28\&60) = 0.0030$, $P(III) = 0.0215$. Values are means and error bars indicate SEM. $*P < 0.05$, $**P < 0.01$, $****P < 0.0001$ (C, D, F, H–J, Student's t tests). Source data are available online for this figure.

the brain serves as a more antecedent and dominant chromatin regulator than CBP, which is responsible for a concomitant function with GAF.

Synergetic binding in multiple GAF-binding sites facilitates the rapid regulation of open chromatins. We found that GAF-binding sites in the promoter region of *henna* are critical for regulating behavioral changes of locusts within a few hours. Due to the presence of multiple predicted GAF-binding sites in the locust genome, we cannot rule out the possibility that GAF modulates the behavioral plasticity of locusts, not solely through DHS1 and *henna*. In addition, differential GAF-binding sites have been detected in environmentally induced plastic phenotypes of diverse insects using genome-wide profiling approaches alone. Although GAF-binding sites are mentioned in differential analyses, the importance of GAF-binding sites is ignored in these studies. For example, GAF-binding sites are enriched in the peaks of differential chromatin openness and in the regulatory regions of differentially expressed genes between specialized layers and foragers in the honey bee (Jones et al, 2020). In regions of differential chromatin openness, GAF-binding sites are also significantly more occupied in worker bees than in queen bees (Lowe et al, 2022). In addition, GAF-binding sites are significantly enriched in promoters of caste-specific genes in the carpenter ant (Simola et al, 2013). Furthermore, in butterfly wing pattern variation, the GAF binding site represents one of the top two enriched motifs in open chromatin regions in the forewing, compared to those in the hindwing (Connahs et al, 2022). However, because the alternative phenotypic forms involved in these studies are well-differentiated developmentally over a period of several days or months, the role of GAF-binding sites in chromatin-based regulation during short-term transitions of phenotypes remains unknown.

In locusts, the GAF expression and the chromatin openness of GAF-binding sites showed rapidly changes in changes of population density for several hours. Three GAF-binding sites in DHS1 serve two regulatory roles, including activating promoter activity and maintaining chromatin openness, indicating a synergetic manner of multiple GAF-binding sites. The close proximity of multiple GAF-binding sites in DHS1 corresponds to GAF multimerization that is mediated by the POZ domain at the N terminus of GAF (Espinás et al, 1999). The loss of GAF60 in DHS1 resulted in the reduction of promoter activity. The mutation of either GAF28 or GAF166 affects the ability to remodel chromatin openness at DHS1. These results suggest that none of the three GAF-binding sites are functionally redundant, each contributing

independently to rapid GAF-dependent chromatin regulation. GAF, which is exclusively found in insects, emerged in the last common ancestor of insects (Pauli et al, 2016). We revealed the widespread GAF-binding sites in the flanking region of *henna* TSSs of several holometabolous and hemimetabolous insects. In particular, evidence for dopamine to regulate phenotypic plasticity has been demonstrated in phase change of locusts, wing plasticity of pea aphid, caste-specific behavior of honey bee, and progenitive plasticity of brown planthopper (Liu et al, 2022; Liu and Brisson, 2023; Sasaki et al, 2018). These results imply that the persistence of multiple GAF-binding sites in the promoter region of *henna* is evolutionarily conserved in dopamine signaling pathway in insects. Therefore, due to the need for quick dopamine level changes in the brain, synergetic binding among multiple GAF-binding sites facilitates the rapid chromatin regulation of *henna* during phenotypic changes over short timescales in insects.

## Methods

### Reagents and tools table

| Reagent/resource | Reference or source | Identifier or catalog number |
|---|---|---|
| **Experimental models** | | |
| Locust (*Locusta migratoria*) | Lab-reared | |
| **Recombinant DNA** | | |
| pGL4.10 | Promega | Cat# E6651 |
| pGL4.10 with the recombinant DHS1 sequence | This study | |
| pGL4.10 with the recombinant DHS2 sequence | This study | |
| PAC5.10/V5-His B vector | Invitrogen | Cat# V411020 |
| PAC5.10/V5-His B with the recombinant GAF sequence | This study | |
| PAC5.10/V5-His B with the reversely recombinant GAF sequence | This study | |

| Reagent/resource | Reference or source | Identifier or catalog number |
|---|---|---|
| **Antibodies** | | |
| Rabbit polyclonal anti-GAF | This study | |
| Rabbit polyclonal anti-PAH | This study | |
| Rabbit polyclonal anti-H3 | Easybio | Cat# BE3222 |
| Mouse polyclonal anti-V5 | Easybio | Cat# BE2033 |
| Goat anti-mouse IgG-HRP | Easybio | Cat# BE0102 |
| Goat anti-rabbit IgG-HRP | Easybio | Cat# BE0101 |
| Rabbit IgG | Millipore | Cat# PP64B |
| Alexa Fluor 546 goat anti-rabbit IgG | Life Technologies | Cat# A-11035 |
| Goat (Fab)-anti-rabbit FITC | Abcam | Cat# ab7050 |
| **Oligonucleotides and other sequence-based reagents** | | |
| Primers used for DNase-qPCR, ChIP-qPCR, CRISPR/Cas9 PCR, qRT-PCR, and RNAi | This study | Appendix Table S1 |
| Primers used for recombinant plasmid construction | This study | Appendix Table S2 |
| **Chemicals, enzymes, and other reagents** | | |
| TRIzol Reagent | Thermo Fisher | Cat# 15596026 |
| DNase I | Roche | Cat# 03724778103 |
| RQ1 DNase | Promega | Cat# M6101 |
| Protease inhibitor cocktail | Thermo Fisher | Cat# 87785 |
| RNasin ribonuclease inhibitor | Promega | Cat# N2111 |
| M-MLV reverse transcriptase | Thermo Fisher | Cat# 28025021 |
| LightCycler 480 SYBR Green I Master Kit | Roche | Cat# 04887352001 |
| SYBR Green Pro Taq HS | Accurate biology | Cat# AG11718 |
| T7 RNA polymerase | Promega | Cat# P2075 |
| Lipofectamine 3000 reagent | Invitrogen | Cat# L3000015 |
| Hochest | Thermo Fisher | Cat# H3570 |
| ApexHF HS DNA Polymerase FS | Accurate biology | Cat# AG12202 |
| Hind III-HF | New England Biolabs | Cat# R3104V |
| Xho I | New England Biolabs | Cat# R0146V |
| BstB I | New England Biolabs | Cat# R0519V |
| Kpn I-HF | New England Biolabs | Cat# R3142V |
| T4 DNA Ligase | New England Biolabs | Cat# M0202V |
| Silk Milk | Easybio | Cat# BE6250 |
| Schneider's Drosophila Medium | Gibco | Cat# 21720024 |
| Dulbecco's modified Eagle's medium (DMEM) | Gibco | Cat# 11965092 |
| Fetal Bovine Serum | Royacel | Cat# RY-F22-05 |

| Reagent/resource | Reference or source | Identifier or catalog number |
|---|---|---|
| Dynabeads His-Tag | Thermo Fisher | Cat# 10103D |
| Cas9 protein | Invitrogen | Cat# A36496 |
| OCT | Sakura Tissue-Tek | Cat# 4583 |
| IGEPAL CA-630 | Sigma | Cat# 18896 |
| Triton X-100 | Solarbio | Cat# T8200 |
| A-amanitin | Sigma | Cat# A2263 |
| 0.5 M EDTA | Invitrogen | Cat# 15575020 |
| Sterilized glycerol | Coolaber Science & Technology | Cat# CG5811-500ml |
| DTT | Invitrogen | Cat# 707265ML |
| Sterilized urea | Coolaber Science & Technology | Cat# CU11631-500g |
| HEPES | Solarbio | Cat# H1095 |
| NP-40 | Solarbio | Cat# N8030 |
| pH 8.0 Tris-HCl | Invitrogen | Cat# AM9855G |
| Imidazole | Coolaber Science & Technology | Cat# CI6381-1kg |
| **Software** | | |
| GraphPad Prism 9 | https://www.graphpad.com/scientific-software/prism/ www.graphpad.com/scientific-software/prism/ | |
| EthoVision XT 13 | Noldus | https://www.noldus.com/ethovision-xt |
| Image-Pro Plus 6.0 | https://mediacy.com/image-pro/ | |
| deepTools v2.27.1 | https://github.com/deeptools/deepTools | |
| Bowtie2 v2.4.1 | https://github.com/BenLangmead/bowtie2 | |
| MACS2 v2.2.7.1 | https://github.com/macs3-project/MACS/wiki/Install-macs2 | |
| **Other** | | |
| Amicon Ultra- 0.5 mL Centrifugal filters | Millipore | Cat# UFC500324 |
| GeneArt Precision gRNA Synthesis Kit | Invitrogen | Cat# A29377 |
| Pierce Magnetic ChIP Kit | Thermo Fisher | Cat# 26157 |
| Chromatin Assembly Kit | Active Motif | Cat# 53500 |
| NEBNext® Ultra™ DNA Library Prep Kit for Illumina® | New England Biolabs | Cat# E7370 |
| NEBNext® Ultra™ II DNA Library Prep Kit for Illumina® | New England Biolabs | Cat# E7645 |
| GeneArt Precision gRNA Synthesis Kit | Invitrogen | Cat# A29377 |
| Illumina Nova X-ten PE150 | Illumina | |

## Insects and treatments

Locusts were fed under standard conditions at the Institute of Zoology, Chinese Academy of Sciences, Beijing, China. The locust colonies were reared under a photoperiod regime of 14 h of light versus 10 h of darkness. The rearing was conducted at a temperature of 30 °C ± 2 °C and fed with fresh wheat seedlings and bran. As gregarious rearing condition, the density of locusts per cage (40 × 40 × 40 cm³) was about 500–600. As a solitary rearing condition, each individual locust was separated in a small cage (10 × 10 × 25 cm³).

For the crowding treatment (CS), every 10 of the fourth-instar solitary nymphs at the first day were reared with 20 of the fourth-instar gregarious nymphs together in a cage (10 × 10 × 10 cm³). For the isolation treatment (IG), the fourth-instar gregarious nymphs at the first day were separately reared under the solitary conditions. For the RNA interference injection, late third-instar nymphs were injected twice over 48-h intervals.

## RNA extraction and RT-qPCR

Total RNA was extracted from insects using TRIzol reagent (Thermo Fisher), and the first strand of cDNA was reverse transcribed with 2 μg of DNase-treated total RNA using oligo dT and M-MLV (Promega). The qRT-PCR was performed using the LightCycler 480 SYBR Green I Master Kit (Roche) on a LightCycler 480 system (Roche). The PCR program was performed with initiation at 95 °C for 10 min, followed by 45 cycles of PCR at 95 °C, 58 °C and 68 °C for 20 s. Relative expression levels were calculated by the $2^{-\Delta\Delta Ct}$ method, normalized to the housekeeping gene ribosomal protein49 (Rp49). Three to six biological replicates were performed for each treatment.

## Chromatin-associated RNA extraction and RT-qPCR

Freshly dissected brains of homozygous DHS-mutated and gregarious locusts were frozen in liquid nitrogen and then homogenized by a Dounce grinder. Nuclei were collected by centrifugation and washed twice in the washing buffer (1 mM EDTA, 0.1% Triton X-100, 1× protease inhibitor mix, 0.025 mM a-amanitin, and RNasin ribonuclease inhibitor in PBS). After the supernatant was removed, nuclei were resuspended in glycerol buffer (20 mM pH 8.0 Tris-HCl, 70 mM NaCl, 0.5 mM EDTA, 50% filter-sterilized glycerol, 0.84 mM DTT, 1× protease inhibitor mix, 0.025 mM a-amanitin, and RNasin ribonuclease inhibitor in RNase-free H2O) and then lysed on ice for 2 min in nuclei lysis buffer (1% NP-40, 20 mM pH7.5 HEPES, 0.2 mM EDTA, 300 mM NaCl, 1 M filter-sterilized urea, 1 mM DTT, 1× protease inhibition mix, 0.025 mM α-amanitin, and RNasin ribonuclease inhibitor in RNase-free H2O). After centrifugation and removal of the supernatant, the collected chromatins were resuspended in chromatin resuspension solution (1× protease inhibition mix, 0.025 mM a-amanitin, and RNasin ribonuclease inhibitor in PBS). The RNA extraction and quantitative PCR analysis were the same as those previously mentioned.

## RNA interference

The dsRNA of target genes was prepared by using the T7 RiboMAX system (Promega) and annealed at 70 °C for 10 min. Fourth-instar nymphs were positioned in a Kopf stereotaxic frame specifically adapted for locust surgery. Using Nevis scissors, a midline incision ~2 mm long was made at the midpoint between the two antennae, exposing the underlying brain. A total of 2 μg of dsRNAs (2 μg/μl) for *henna* TSS1, TSS2, and GAF were injected into the brain using a glass micropipette tip mounted on a nanoliter injector (World Precision Instruments) under an anatomical lens. The dsRNAs of GFP were used as the negative controls. The injected locusts were returned to their normal rearing conditions and kept for 48 h before their brains were harvested for RNA or protein extraction, or for behavior assays. We used randomization to assign individual locusts to the dsRNA groups to avoid selection bias and blinded the observer to the dsRNA groups during data collection and analysis to reduce subjective bias. The migratory locust possesses a highly sensitive and systemic RNAi response, capable of spreading expression silencing from the injection sites to distant cells in the brain (Luo et al, 2013).

## DNase-seq and DNase-qPCR

Fourth-instar brain was crosslinked in 1% formaldehyde for 10 min at room temperature, and crosslinking was stopped by the addition of glycine. The isolated and suspended nucleus was then collected after tissue mincing in ice-cold PBS. The nuclear suspension was lysed in 0.1% IGEPAL-630 (SIGMA) for 15 min on ice. The suspension was digested with DNase I (Roche, 03724778103) at 0.1 U per 100 μl in all for 10 min. Total digested DNA was precipitated with ethanol when protein was digested with Proteinase K. In total, 100 ng of precipitated DNA was used to construct Illumina sequencing libraries according to protocols (NEB, E7370). The libraries were sequenced using Illumine Hiseq X-ten and Nova. To construct templates for DNase-qPCR, the 100 ng of digested DNA was ligated with an Illumina sequencing adapter and pre-PCR for 6–8 cycles with sequencing primers. A 10 ng pre-PCR mix was used for relative quantity PCR with specific primers that bind to *henna* DHS1 and DHS2 regions, *rp49* DHS regions, and inaccessible regions. The length of all PCR products ranged from 100 to 200 bp.

## Processing of DNase-seq data

The raw sequencing reads of DNase-seq sequencing were processed with Trim Galore v0.6.5 for adapter removal and quality filtering. The trimmed reads were aligned to the locust genome using Bowtie2 v2.4.1 (Langmead and Salzberg, 2012). The DNase-seq peaks were called on de-duplicated reads by MACS2 v2.2.7.1 using the narrow peak parameters without the --broad and --broad-cutoff options (Feng et al, 2012). The DNase-seq signals were enriched at TSSs that were identified by Oligo-capping sequencing in a previous study (Liu et al, 2021). The metaplot enrichment plots were generated using deepTools v2.27.1 (Ramírez et al, 2014).

## CRISPR/Cas9-mediated fragment mutagenesis

The guide RNAs (gRNAs) were designed using the CasOT software to minimize potential off-target sites (Xiao et al, 2014). The gRNA (DHS1-gRNA) contains a 20-base target sequence immediately upstream of the PAM. The resulting gRNAs were first assessed for cleavage efficiency individually, and then two neighboring gRNAs were selected for further fragment mutagenesis experiments. The gRNA targeting the *henna* DHS1 region was synthesized using the

GeneArt Precision gRNA Synthesis Kit (Invitrogen, A29377). The assembled DNA templates, containing the T7 promoter and the gRNA sequence, were subjected to in vitro transcription using the TranscriptAid Enzyme Mix. The DNA templates were then removed by DNase I digestion immediately after in vitro transcription. The transcribed gRNAs were purified using the gRNA Clean-Up Kit. The donor sequence was a 150 bp sequence without core promoter motifs (such as Inr or TATA box) (primer Inserted seq). The insertion of the donor sequence resulted in the dysfunction of DHS1 by expelling DHS1 from the transcription start site. The donor DNAs (GenScript), which contained 150-bp homologous arms, were modified with C6-PEG10 at the 5' end. The freshly oviposited embryos, deposited in sands within 2 h, were collected from egg pods and washed with a series of 70% ethanol and deionized water. A 13.8-μl mixture of Cas9 protein (Invitrogen, A36496, 300 ng/μl), donor DNA (300 ng/μl), and gRNA (150 ng/μl) was injected a nanoliter injector (World Precision Instruments) into the top end of the eggs, where germ cells are located. The treated eggs were incubated at 30 °C on 1% agarose gel. The tarsal ends of the hinge legs were scissored to extract DNA to quantify the genotypes (primer Hn-insertest-2). The mRNA and protein levels of *henna* were then quantified in brains and fat bodies.

## Behavioral assays

Behavioral assays were conducted in a rectangular arena $(40 \times 30 \times 10 \text{ cm}^3)$ as in previous research (Guo et al, 2011). The EthoVision system (Noldus) was used for data acquisition and data analysis. A binary logistic regression model approach was used (Yang et al, 2014): Pgreg = en /(1+en), where $n = \beta0 + \beta1 \times 1 + \beta2 \times 2 + \beta = \beta kXk$, and where Pgreg indicated the probability of a locust being gregarious. Pgreg = 1 indicated fully gregarious behavior and Pgreg = 0 indicated fully solitary behavior.

## Reporter and expression plasmid construction and dual-luciferase reporter assay

The *henna* DHS1 and DHS2 regions were individually ligated into the pGL4.10 vector (Promega, E6651) by using the Hind III and Xho I restriction sites. The sequences of the plasmids were confirmed by Sanger sequencing. The full length of the GAF coding sequence was cloned into PAC5.10/V5-His B vector (Invitrogen, V411020) by using the BstB I and Kpn I restriction sites. For the mutated GAF-binding sites, the core motifs "GAGAG" and "CTCTC" (Adkins et al, 2006), were replaced with by "AAAAAA".

These plasmids were co-transfected into *Drosophila* S2 cells (ATCC, Cat#CRL-1963, RRID: CVCL_Z232) plated on 96-well plates using Lipo3000 (Invitrogen, L3000015) and the plasmid pGL4.73 (Promega) was as an internal reference. For GAF overexpression, S2 cells were co-transfected with the expression plasmid. A total of 100 ng plasmid in all was co-transfected into each well, and 2 ng pGL4.73 was used as an internal reference. Luciferase intensity was measured using a dual-luciferase reporter assay system (Promega, E1910) 48 h after incubation.

## Immunofluorescence and antibodies

The polyclonal antibodies for PAH and GAF were produced by immunizing rabbits with prokaryotic expression antigen (Beijing Protein Innovation). Polyclonal antibodies for H3 (BE3222), V5 (BE2033) were commercial products (Beijing Easybio). An immunofluorescence experiment was performed as previously described with some modifications (Yang et al, 2014; Zhang et al, 2020). A combined in situ analysis of the proteins PAH and GAF was conducted in the brains of 4th-instar nymphs of gregarious locusts. Brains were dissected under an anatomical lens and fixed in 4% paraformaldehyde overnight. Brain sections (5 μm thick), embedded in OCT (Sakura Tissue-Tek), were then washed twice in PBS for 5 min each. The sections were incubated in PBST (0.2% Triton X-100 in PBS) for 20 min, followed by two washes in PBS for 5 min each. The sections were then incubated in 5% bovine serum albumin in PBST at 25 °C for 60 min, and then incubated with the polyclonal antibody GAF (1:50) at 37 °C for 60 min, followed by three washes with PBS. The slides were then incubated with goat (Fab)-anti-rabbit FITC (1:500, Abcam, ab7050) at 37 °C for 30 min. After being washed three times, the slides were incubated with the polyclonal antibody PAH (1:100) at 4 °C overnight. After washing, the slides were incubated with goat anti-rabbit Alexa Fluor 546 (1:1000, Life Technologies, A-11035) at 37 °C for 30 min. The target signals were detected using an LSM 719 confocal fluorescence microscope (Zeiss).

## ChIP-qPCR

The ChIP-qPCR was conducted following the manufacturer's protocol using a Pierce Magnetic ChIP Kit (Thermo, 26157). Freshly collected brains were crosslinked with 1% formaldehyde and washed with ice-cold PBS. Brains were then homogenized to nucleus suspension. The crosslinked nucleus was lysis and digested by MNase. The digested suspension was evenly separated and immunoprecipitated with Normal Rabbit IgG or Rabbit-anti-GAF overnight at 4 °C with rotation. The antibody–protein–DNA complex was enriched by incubation with magnetic protein A/G beads. After the reversal of crosslinking, recovered DNA elution was proceeded to qPCR detection (TIANGEN).

## In vitro chromatin assembly

The in vitro chromatin assembly was conducted following the manufacturer's protocol using a Chromatin Assembly Kit (Active Motif, 53500). Briefly, the Plasmid pGL4.10/DHS1 and other plasmids with mutant GAF-binding sites were prepared as sample DNA. The chromatin assembly was an ATP-dependent assay with chaperone protein and ATP-utilizing chromatin assembly and remodeling factor (ACF). In all, 1 μg sample DNA was prepared for assembly for 2.5 h, following the addition of purified locust GAF/His. After the addition of GAF, the sample was incubated for assembly for another 2.5 h. The analysis of assembled chromatin was revealed by partial digestion for 2 min. At least 200 ng chromatin was visualized by agarose gel electrophoresis.

The purification of expressed GAF/His was by the Dynabeads His-Tag Isolation and Pulldown (Thermo Fisher, 10103D). A total protein from S2 cell and expressed GAF/His was added to the Dynabeads, incubating for 24 h. The purified GAF/His was eluted by 300 mM Imidazole. Finally, the buffer exchanged into PBS using Amicon Ultra- 0.5 mL Centrifugal filters (Millipore, UFC500324) following the manufacturer's protocol. The quality of purified protein was visualized by western blots.

## Primers

All primers used for DNase-qPCR, ChIP-qPCR, CRISPR/Cas9 PCR, qRT-PCR, and RNAi are listed in Appendix Table S1. Primers used for recombinant plasmid construction are listed in Appendix Table S2.

## Statistical analyses

All data are shown as the means ± SEMs of at least three biologically independent experiments. The statistical significance of differences between treatments was mostly analyzed using Student's *t* test or one-way analysis of variance (ANOVA) followed by Tukey's test for multiple comparisons and a $P = 0.05$ significance threshold level. Data were analyzed using GraphPad Prism 9. Protein intensity was quantified using Image-Pro Plus. Graphs were generated using GraphPad Prism 9.

# Data availability

All data are available in the main text or the supplementary materials. The sequencing data have been deposited in the Science Data Bank (ScienceDB) Database under the DOI accession: https://doi.org/10.57760/sciencedb.18341.

The source data of this paper are collected in the following database record: biostudies:S-SCDT-10_1038-S44318-025-00428-x.

# Peer review information

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

## Acknowledgements

This research was supported by grants from the Ministry of Agriculture and Rural Affairs of China, the National Natural Science Foundation of China (32088102 and 32300397), the Initiative Scientific Research Program (Institute of Zoology, Chinese Academy of Sciences, No. 2024IOZ0202), and the National Key R&D Program of China (No. 2022YFD1400500). The computational resources were provided by the Research Network of Computational Biology and the Supercomputing Center at the Institute of Zoology, Chinese Academy of Sciences.

## Author contributions

**Xiao Li**: Validation; Investigation; Methodology; Writing—original draft. **Feng Jiang**: Conceptualization; Formal analysis; Supervision; Funding acquisition; Writing—original draft. **Qing Liu**: Formal analysis; Funding acquisition; Methodology. **Zizheng Zhang**: Investigation. **Wenjing Fang**: Investigation. **Yutong Wang**: Investigation. **Hongran Liu**: Formal analysis. **Le Kang**: Conceptualization; Funding acquisition; Writing—review and editing.

Source data underlying figure panels in this paper may have individual authorship assigned. Where available, figure panel/source data authorship is listed in the following database record: biostudies:S-SCDT-10_1038-S44318-025-00428-x.

## Disclosure and competing interests statement

The authors declare no competing interests. The funders had no role in the study design, data collection, analysis, decision to publish, or preparation of the manuscript.

# Expanded View Figures

**A**

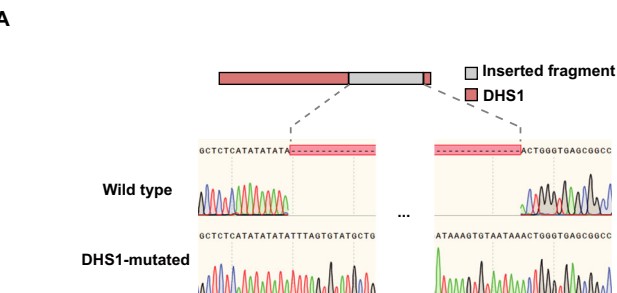

**B**

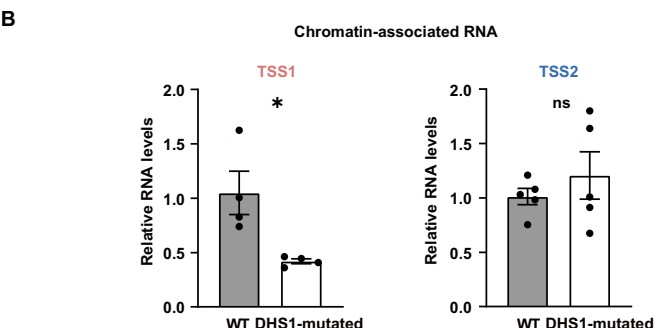

**Figure EV1. The chromatin-associated RNA level of TSS1 and TSS2 in the brains of homozygous DHS1-dysfunctional mutants of gregarious locusts.**

(**A**) A diagram of the gene structure of homozygous DHS1-dysfunctional mutants of gregarious locusts. (**B**) The chromatin-associated RNA level of TSS1 and TSS2 in the brains of homozygous DHS1-dysfunctional mutants of gregarious locusts. Primers were designed specifically for the TSS. N = 4 for TSS1, and n = 5 for TSS2. WT, wild type. *P* values from left to right, $P = 0.0286$ (Mann–Whitney test), $P = 0.4221$. Values are means and error bars indicate SEM. *$P < 0.05$, Student's t tests. Source data are available online for this figure.

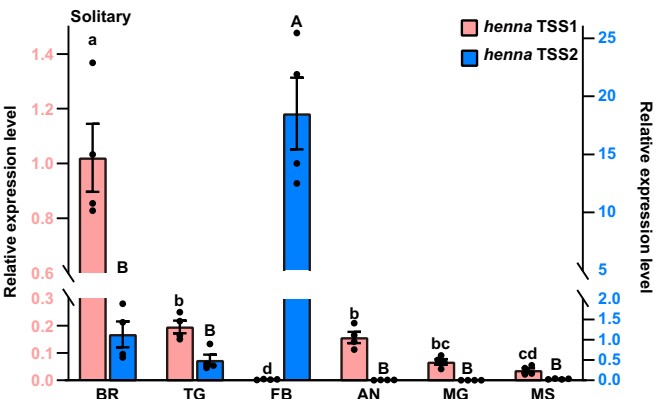

**Figure EV2. The mRNA expression pattern for henna TSS1 and TSS2 in solitary locusts.**

$N = 4$ for each group. Values are means and error bars indicate SEM. $P = 0.0020$ for TSS1 (Brown–Forsythe ANOVA test), $P < 0.0001$ for TSS2 (ordinary one-way ANOVA tests). BR, Brain. TG, thoracic ganglia. FB, fat body. AN, antenna. MG, Midgut. MS, muscle. Same letters above the bars indicate no significant difference between groups. Different letters indicate a significant difference between groups. Combined letters indicate groups are different from others but not from each other. Source data are available online for this figure.

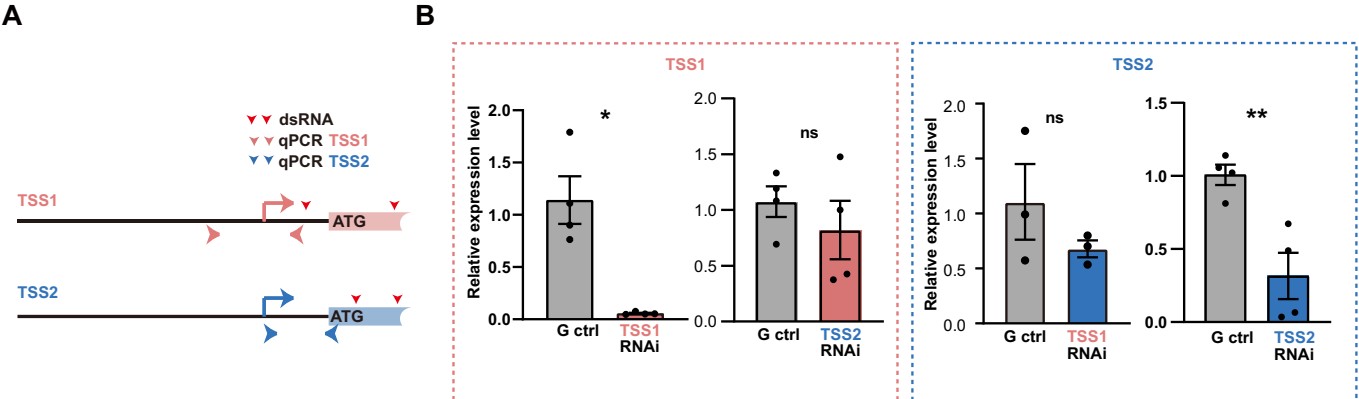

**Figure EV3.  The reduction efficiency of the henna TSS1 and TSS2 was independent.**

(**A**) A schematic diagram of the location of primers for RNAi and qPCR. (**B**) In red dotted box, it was the expression of TSS1 for gregarious control (G ctrl), TSS1 RNAi and TSS2 RNAi ($n = 4$ for TSS1 groups, G ctrl and TSS2 RNAi of TSS2 groups; $n = 3$ for G ctrl and TSS1 RNAi of TSS2 groups). Of TSS1, $P$ values from left to right, $P = 0.0286$, $P = 0.4213$. Of TSS2, $P$ values from left to right, $P = 0.2932$, $P = 0.0071$. Values are means and error bars indicate SEM. *$P < 0.05$, **$P < 0.01$. Of TSS1, G ctrl versus TSS2 RNAi, and groups of TSS2, Student's $t$ tests. Of TSS1, G ctrl versus TSS1 RNAi, Mann–Whitney test. Source data are available online for this figure.

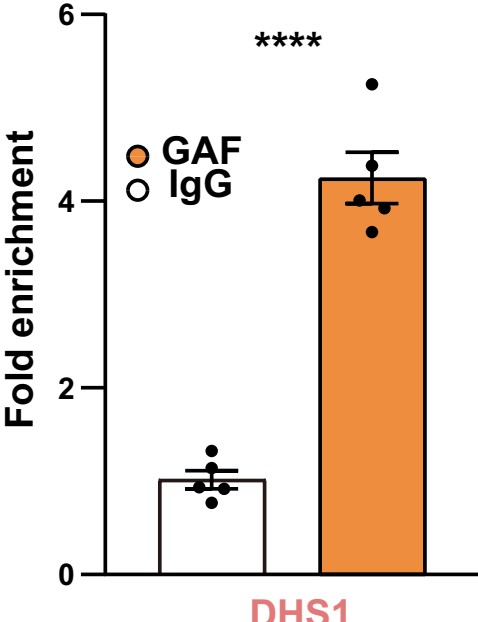

**Figure EV4.   The fold enrichment of GAF binding at DHS1 in vivo.**

$N = 5$ for each group. $P = 4.02\text{e-}6$. Values are means and error bars indicate SEM. ****$P < 0.0001$, Student's *t* tests. Source data are available online for this figure.

