## [Peer Review File · The EMBO Journal]

GAF-dependent Chromatin Plasticity Determines Promoter Usage to Mediate Locust Gregarious Behavior

Xiao Li, Feng Jiang, Qing Liu, Zizheng Zhang, Wenjing Fang, Yutong Wang, Hongran Liu, and Le Kang

Corresponding author(s): Le Kang (lkang@ioz.ac.cn)

Review Timeline:

Submission Date:	13th Jun 24
Editorial Decision:	21st Sep 24
Revision Received:	24th Dec 24
Editorial Decision:	11th Feb 25
Revision Received:	17th Feb 25
Accepted:	22nd Feb 25

Editor: Yehu Moran

Transaction Report:

Dear Dr. Kang,

Thank you for submitting your manuscript for consideration by the EMBO Journal.

I would like to start by apologizing for the long time it took to get back to you with a decision. This stems from the fact that unfortunately an expert who agreed to review your manuscript stopped communicating at some point and we had to replace them.

Anyway, the manuscript has now been seen by three referees whose comments are shown below.

Given the referees' positive recommendations, I would like to invite you to submit a revised version of the manuscript, thoroughly addressing the comments of all three reviewers. I should add that it is EMBO Journal policy to allow only a single round of revision, and acceptance of your manuscript will therefore depend on the completeness of your responses in this revised version.

As some referee comments were quite substantial, I would also like to suggest that before you start your revision process, you will send us a list of planned major revision steps so we can agree on a revision plan in advance. From our experience this can greatly increase the chances of eventually accepting your manuscript after revision.

Thank you for the opportunity to consider your work for publication. I look forward to your revision.

Yours sincerely,

Yehu Moran
Academic Editor
The EMBO Journal

We realize that it is difficult to revise to a specific deadline. In the interest of protecting the conceptual advance provided by the work, we recommend a revision within 3 months (20th Dec 2024). Please discuss the revision progress ahead of this time with the editor if you require more time to complete the revisions. Use the link below to submit your revision:

Referee #1:

The manuscript entitled "GAF-1 dependent Chromatin Plasticity Determines Promoter Usage to Mediate Locust Gregarious Behavior" by Li et al. describes the role of GAF-dependent regulation of henna transcription in locust behavioral polyphenism. The manuscript includes detailed characterization of the henna promoter that is responsible for the regulation of behavior by henna via the action of GAF. Establishing causal links between genetic factors and complex behavioral phenotypes in organisms that are not genetically tractable is not trivial. Therefore, this manuscript could represent an important proof-of-principle. However, in its current form, the manuscript suffers from several weaknesses that reduce its potential impact. Below are more specific comments that I hope the authors will find useful.

1. As written, this manuscript is very hard to follow. The writing is dense, imprecise, and includes many careless typos and incomplete sentences. The manuscript will benefit tremendously from professional editing for language and structure.
2. The introduction is too long. Many of the examples of insect phenotypic polyphenisms covered have no direct relevance to the locust phenomena.
3. Many of the methods are poorly described, and do not include enough information to allow evaluation of approach, and experimental outcomes. For example, no information is provided for how GAF binding sites were identified bioinformatically, and which other TF binding sites were identified. Furthermore, no control genomic regions were used to demonstrate that GAF binding is enriched henna promoter regions. Other important missing details relate to the parameters and methods used for the design of the dsRNA and CRISPR reagents.
4. Although the authors used whole genome approach for identifying potential loci associated with transcriptional regulation, they then use a gene candidate approach to choose henna as their primary candidate for follow-up studies. If henna is the best candidate, then why not focusing on this gene from the beginning. Alternatively, if other loci seem as promising, why not describing these in more details. As written, data related to chromatin oneness across the whole genome could be removed because their relevance to the main thesis of the study is marginal.
5. To my knowledge, this is the first report of a successful homology-dependent CRISPR-cas9 genome editing in the germline of any grasshopper species. Since so many others in the field have failed, including detailed protocol for how this was achieved, and providing molecular data that support successful germline transformation are required to increase confidence in the reported data.
6. Brain injections of dsRNA are notoriously difficult because of the impact of the injection-related injuries on behavior. The authors should include detailed protocols for how they achieved successful dsRNA injections, how did they choose which animals to include in their analysis, and what approaches were used to ensure precise targeting of specific brain regions. Including brain sections that show injection site would increase confidence in the included data.
7. Imaging data shown in Fig. 4L are oversaturated and noninformative.
8. Because the orthologs of GAF and Grh are not currently annotated in the genome of *Locusta migratoria* genome, providing the specific LOC number of each identified gene is important.

Minor comments:

1. Phenotypic plasticity is not defined by the ability of a single genome to produce distinct phenotypes. In the case of locust, the two different phases should be defined as "behavioral polyphenism", which is the capacity of the genome to encode two alternative behavioral states.
2. It is not clear what the sentence "Selection of henna promoter usage determines the playground of PAH enzymes" means (line 403).
3. The relevance of PAH phosphorylation states in humans to the discussion of the data presented here is not obvious (lines 409-418).

4. It is not clear why the authors claim that "Synergetic binding in multiple GAF binding sites facilitates the rapid regulation of open chromatin" (lines 464-465). I thought that the model is that once chromatin is open, synergy between enhancer elements can regulate expression. Unless I have missed it, the presented data do not show that GAF binding of DNA regulates chromatin openness.

Referee #2:

In their work, Li, Jiang, and colleagues investigate the epigenetic basis for gene expression and behavior in the migratory locust *Locusta migratoria*. More specifically, they dissect pre-transcriptional regulation of the gene *henna* via GAGA factor (GAF) mediated changes in chromatin conformation at cis-regulatory sites. Through a series of highly detailed genomic, in vitro, in vivo, and behavioral experiments, their work demonstrates GAF binding to promoter regions of *henna* operates in a tissue specific manner to modulate gene expression associated with behavioral phase changes in locusts. Overall, this is an impressive body of work that integrates innovative and highly detailed methodology (especially by non-model system standards) and makes significant conceptual advances in determining the mechanistic basis by which chromatin conformation modulates gene expression and insect behavior, a research area of much needed attention. While I have a few suggestions on the presentation of the results, the findings are laid out in a highly logical and elegant fashion that makes understanding the results very accessible. Publication of this work will provide significant conceptual and methodological advances in the field of behavioral plasticity, gene regulation, and perhaps in applied settings such as pest management. I have some major but primarily minor comments (presented chronologically) that I hope will improve the clarity of the results' presentation and interpretation.

Major comments:

- 1) A crucial part of many experiments focuses on the 4 hour exposure period of crowdedness or solitude to induce a behavior response. I think it would be helpful to explain or cite how this 4 hour period was determined because to a non-locust researcher this period of time may seem arbitrary.
- 2) Lines 349-351/ Figure 6F: The authors state increasing GAF resulted in a gradual decrease in tri-nucleosome occupancy. However, this result is not quantified and the figure panel referenced (6F), to me, does not show any detectable decrease in tri-nucleosome occupancy. Lanes 1-3 do not look particularly dynamic or different and only lane 4 (highest GAF amount) is a qualitatively different bad pattern shown. If I'm interpreting this correctly, I would ask the authors to amend their writing to reflect this distinction, or lack thereof, between lanes as I'm not sure a dose-dependent relationship is shown in 6F. If I'm incorrectly interpreting this, please provide some sort of quantification to demonstrate tri-nucleosome occupancy is decreasing as GAF concentration increases. Furthermore, please specify the range GAF quantity used in the pink ramp-up part of the figure.

Minor comments:

A) Introduction

- 1) Lines 97-98: well said

B) Results

- 2) Line 117: "open chromatin"
- 3) Line 122: switch order of enrichment and upstream.
- 4) Lines 135-137: The authors state H3K36me3 deposition starts at DHS1 and spreads to coding sequence. However, examining Figure 1 shows a vanishingly small peak of H3K36me3 at DHS1 that looks little more than noise. Perhaps stating this mark occurs downstream of DHS1 is more accurate.
- 5) Line 137: "suggested" not "demonstrated"
- 6) Lines 170: "Ho TSS3" is confusing, not sure what "Ho" is referring to
- 7) Line 283: "sites" typo
- 8) Line 284: "precluded" perhaps too strong. "inhibited" is more appropriate

C) Discussion

- 9) Line 408: please specify what percent identity defines a "large majority"
- 10) Lines 431-432: please cite relevant work for this claim'
- 11) Is there a reason why aggregation pheromone wasn't applied in this study? Given the behavioral work it isn't necessary, but could be a nice complement to the molecular and organismal results
- 12) Line 460-463: This statement is claiming too much in my opinion by comparing the relative contribution of GAF to CBP in chromatin regulation, as this study did not assess CBP in locusts. Please consider revising
- 13) Line 484: typo "GAT"

D) Methods

- 14) Lines 558-559: please specify what "narrow parameters" were used in MACS

E) Figures

- 15) I'd highly recommend switching the order of panels in Figure 1F to that the DHS1 result is on top and DHS2 result is on

bottom. This will align better with the brain vs fat body panels presented in Figure 1E

16) Throughout the figures "a" "b" "bc" and "c" are noted on top on bar graphs. Please specify what these mean in the legend

Reservation: My background in insect behavioral studies is lacking, so I have refrained from providing feedback on the experimental behavior results

Referee #3:

General summary

The undertaken study is tackling how short-term epigenetic regulation might account for the gregarious or solitary behavior of locusts upon response to environmental changes, which defines a challenge of interest for crossing scales from behavior and phenotypic plasticity down to molecular levels regulating an important regulator of dopamine involved in gregarious locusts, namely henna. In particular, the authors identify a putative enhancer that is differentially accessible upon such environmental change. They show that among the possible TFs involved, GAF is a bona fide candidate. GAF actually mediates chromatin accessibility of the corresponding enhancer and is able to drive the expression of henna.

While it is not surprising that chromatin accessibility is linked to the specific changes of chromatin associated with developmental cues and enhancers, the authors still characterize a complex system from behavior involving a central gene in the process down to molecular scales involving GAF. An ideal demonstration would be to demonstrate that without GAF, locusts are truly unable to modulate their collective behavior though the outcome of such an experiment can be obviously complex.

Major points:

1. Fig 1: It is not clear how many genes harbor several DHS in their bodies, and if Henna is in that sense particular. Furthermore, the brief analysis of histone marks is of correlative nature, and so the authors can hardly claim that: « These chromatin characteristics demonstrated that DHS1 was the promoter region that initiated the predominant transcription of henna in the brains of gregarious locusts. » There is nothing like a demonstration in the finding that DHS1 is the key element to initiate predominant transcription. This is more a matter of transcriptional induction, not simply of chromatin accessibility.
2. To assess the influence of DHS1 on transcription per se, the authors may probe nascent RNAs instead of mRNAs that also depends on RNA stability (Figure 1E). Given the position of DHS1 within an intron, this could be another mechanism of action that cannot be excluded.
3. While the presence of GAF at TSS1-unlike TSS2- is of interest, it should be tested whether its binding is impaired in the mutant DHS1.
4. Although other factors might be involved, the flow of arguments involving GAF binding to DHS1 is documented. However, ATAC-seq would have been more convincing as a way to re-scan the whole pattern of DHS in the henna gene locus, depending on presence absence of GAF in the cells.
5. There is not direct demonstration that GAF is truly required to drive the change of behavior of locusts, though it is shown to be involved in DHS1-mediated expression of henna, which is required to such collective behavior (Fi. 3A). Re-assessing the gregarious behavior of locusts upon GAF depletion or after CRISPR of GAF sites from DHS1 would be needed to reach such a conclusion.

Minor points:

1. I'm not sure of the terms « wild type compared to gregarious locusts », if gregarious is a behavior that is context dependent (but has the same genotype)?
2. Figures were not numbered.
3. In Fig 6b, a control is needed to test if GAF depletion is specifically altering DHS1 as opposed to another site or locus as every measure shows a decrease of H3.

Dear Professor Moran,

On behalf of my co-authors, I sincerely thank you for giving us the opportunity to revise our manuscript (Li et al., Ms. No. EMBOJ-2024-118181R). We greatly appreciate you and the reviewers for their positive and constructive suggestions on our manuscript. The suggestions greatly improve the quality and presentation of this manuscript. We have made our best efforts to complete the supplementary experiments, revise our manuscript according to the comments, and answer all the questions from the three reviewers point by point.

The major revisions include:

1. We quantified the nascent RNA and protein levels of *Henna* in the brains of gregarious locusts.
2. The dual immunofluorescence image of GAF and *Henna* in the brains of gregarious locusts was re-provided.
3. We performed a quantitative analysis of di-nucleosome occupancy that decreases as GAF concentration increases.
4. The results of an EMSA experiment are added to verify the impaired interaction between GAF and the mutant DHS1 in vitro.

Please find attached the revised version, which we would like to submit for your kind consideration. We would like to express our sincere appreciation to you and the reviewers for your comments on our paper. Looking forward to hearing from you.

Best regards,

Le Kang, Ph.D.

CAS Distinguished Professor

Institute of Zoology

Chinese Academy of Sciences (CAS)

Beijing 100101, China

Tel: 86-10-6480-7219

Fax: 86-10-6480-7099

E-mail: lkang@ioz.ac.cn

Response to the Reviewers

Explanation: All editorial correspondence from *EMBO J*, including the reviewers' comments, is verbatim in black. Our responses are inserted directly into this text in blue. All changes have been implemented in the final version of the manuscript.

Dear Dr. Kang,

Thank you for submitting your manuscript for consideration by the EMBO Journal.

I would like to start by apologizing for the long time it took to get back to you with a decision. This stems from the fact that unfortunately an expert who agreed to review your manuscript stopped communicating at some point and we had to replace them.

Anyway, the manuscript has now been seen by three referees whose comments are shown below.

Given the referees' positive recommendations, I would like to invite you to submit a revised version of the manuscript, thoroughly addressing the comments of all three reviewers. I should add that it is EMBO Journal policy to allow only a single round of revision, and acceptance of your manuscript will therefore depend on the completeness of your responses in this revised version.

When preparing your letter of response to the referees' comments, please bear in mind that this will form part of the Review Process File, and will therefore be available online to the community. For more details on our Transparent Editorial Process, please visit our website:

<https://www.embopress.org/page/journal/14602075/authorguide#transparentprocess>

As some referee comments were quite substantial, I would also like to suggest that

before you start your revision process, you will send us a list of planned major revision steps so we can agree on a revision plan in advance. From our experience this can greatly increase the chances of eventually accepting your manuscript after revision.

Thank you for the opportunity to consider your work for publication. I look forward to your revision.

Yours sincerely,

Yehu Moran

Academic Editor

The EMBO Journal

Referee #1:

The manuscript entitled "GAF-1 dependent Chromatin Plasticity Determines Promoter Usage to Mediate Locust Gregarious Behavior" by Li et al. describes the role of GAF-dependent regulation of henna transcription in locust behavioral polyphenism. The manuscript includes detailed characterization of the henna promoter that is responsible for the regulation of behavior by henna via the action of GAF. Establishing causal links between genetic factors and complex behavioral phenotypes in organisms that are not genetically tractable is not trivial. Therefore, this manuscript could represent an important proof-of-principle. However, in its current form, the manuscript suffers from several weaknesses that reduce its potential impact. Below are more specific comments that I hope the authors will find useful.

1. As written, this manuscript is very hard to follow. The writing is dense, imprecise, and includes many careless typos and incomplete sentences. The manuscript will benefit tremendously from professional editing for language and structure.

Response #1: Thanks for your suggestion. We carefully rechecked our manuscript to correct imprecise descriptions, types, and incomplete sentences. We also invited a native-speaking expert to polish the manuscript.

2. The introduction is too long. Many of the examples of insect phenotypic polyphenisms covered have no direct relevance to the locust phenomena.

Response 2: Thank you for your suggestion. We removed some sentences that were not related to locust polyphenism in the revision. Now, only the case for honey bees is presented because it is relevant to TF-directed chromatin structure remodeling, which is also related to the topic of our study.

We deleted the sentences in the revised version:

“In horned dung beetles, the modulation of chromatin openness plays a critical role in specifying nutritional and sexual horn dimorphisms (20). Furthermore, the acquisition of lineage-specific regulatory elements within open chromatin regions represents a genomic innovation in the evolution of nutrition-responsive development. In pharaoh ants, open chromatin regions near caste brain-specific genes are associated with social function of queens, males, and workers (21).”

3. Many of the methods are poorly described, and do not include enough information to allow evaluation of approach, and experimental outcomes. For example, no information is provided for how GAF binding sites were identified bioinformatically, and which other TF binding sites were identified. Furthermore, no control genomic regions were used to demonstrate that GAF binding is enriched henna promoter regions. Other important missing details relate to the parameters and methods used for the design of the dsRNA and CRISPR reagents.

Response #3: We appreciate your comments. In our previous version, we described the methods for identifying TFBS in the legend of Fig. EV6. The insect core database, which includes GAF binding sites, was downloaded from the JASPAR website (Castro-Mondragon et al., *Nucleic Acids Res*, 2022). FIMO (Find Individual Motif Occurrences), part of the MEME Suite, was then employed to scan genomic sequences for significant matches to TFBS motifs (Bailey et al., *Nucleic Acids Res*, 2009). The open chromatin regions were input as regions of interest. Unlike STREME, FIMO does not require control sequences. Background letter frequencies in MEME files were changed to the genomic frequencies of the locust genome.

For RNAi, the primers for dsRNA, which were selected using Primer3, are provided in the Table EV1 of our revised version. For CRISPR, CasOT, a genome-wide Cas9/gRNA off-target searching tool, was used in the design of gRNAs (Xiao et al., Bioinformatics, 2014). The reagents for dsRNAs and CRISPR were described in the RNA interference and CRISPR/Cas9-mediated fragment mutagenesis sections of our previous submission, respectively. We first assessed the cleavage efficiency of an individual gRNA, then selected two neighboring RNAs for further fragment mutagenesis experiments. For more details, please see our response in Response #5.

To add more detail, the sentences were revised as follows:

“The insect core database, which includes GAF binding sites, was downloaded from the JASPAR website, part of the MEME Suite, was then employed to scan genomic sequences for significant matches to TFBS motifs. The open chromatin regions were input as regions of interest. Background letter frequencies in MEME files were changed to the genomic frequencies of the locust genome.”

“The guide RNAs (gRNAs) were designed using the CasOT software to minimize potential off-target sites (48). The gRNA contains a 20-base target sequence immediately upstream of the PAM. The resulting gRNAs were first assessed for cleavage efficiency individually, and then two neighboring gRNAs were selected for further fragment mutagenesis experiments.”

To improve clarity,

4. Although the authors used whole genome approach for identifying potential loci associated with transcriptional regulation, they then use a gene candidate approach to choose henna as their primary candidate for follow-up studies. If henna is the best candidate, then why not focusing on this gene from the beginning. Alternatively, if other loci seem as promising, why not describing these in more details. As written, data related to chromatin oneness across the whole genome could be removed because their relevance to the main thesis of the study is marginal.

Response #4: Thanks for your comments and suggestions. In general, open chromatin regions, where nucleosomes are less densely packed, allowing for binding by transcription factors, are usually several hundred base pairs in length. Therefore, due to the huge genome size of the locust (6.9 G), it is very difficult to identify the exact genomic locations of open chromatin regions without high-throughput screening. This is why developing the DNase-seq method to probe open chromatin regions, or other methods with the same purpose, is crucial in epigenetics. In fact, the gene body region of *Henna* is 88.7 Kb, as shown in Figure 1B. Therefore, performing a genome-wide screening of open chromatin regions using DNase-seq sequencing is the first and very important step in our study. To focus on the main topic of this study, we only include one subfigure for the DNase-seq sequencing to demonstrate the reliability of our experiments. The genome-wide analysis of DNase-seq signals is not directly relevant to the main topic and thus falls outside the scope of this study.

5. To my knowledge, this is the first report of a successful homology-dependent CRISPR-cas9 genome editing in the germline of any grasshopper species. Since so many others in the field have failed, including detailed protocol for how this was achieved, and providing molecular data that support successful germline transformation are required to increase confidence in the reported data.

Response #5: Thanks for this suggestion. We rewrote the method for CRISPR/Cas9-mediated fragment mutagenesis as follows to improve clarity:

“The guide RNAs (gRNAs) were designed using the CasOT software to minimize potential off-target sites (48). The gRNA contains a 20-base target sequence immediately upstream of the PAM. The resulting gRNAs were first assessed for cleavage efficiency individually, and then two neighboring gRNAs were selected for further fragment mutagenesis experiments. The gRNA targeting the *henna* DHS1 region was synthesized using the GeneArt Precision gRNA Synthesis Kit (Invitrogen). The assembled DNA templates, containing the T7 promoter and the gRNA sequence, were subjected to *in vitro* transcription using the TranscriptAid Enzyme Mix. The DNA templates were then removed by DNase I digestion immediately after *in vitro*

transcription. The transcribed gRNAs were purified using the gRNA Clean-Up Kit. The donor sequence was a 150 bp sequence without core promoter motifs (such as Inr or TATA box). The insertion of the donor sequence resulted in the dysfunction of DHS1 by expelling DHS1 from the transcription start site. The donor DNAs (GenScript), which contained 150-bp homologous arms, were modified with C6-PEG10 at the 5' end. The freshly oviposited embryos, deposited in sands within 2 hours, were collected from egg pods and washed with a series of 70% ethanol and deionized water. A 13.8- μ l mixture of Cas9 protein (Invitrogen, A36496, 300 ng/ μ l), donor DNA (300 ng/ μ l), and gRNA (150 ng/ μ l) was injected using a nanoliter injector (World Precision Instruments) into the top end of the eggs, where germ cells are located. The treated eggs were incubated at 30°C on 1% agarose gel. The tarsal ends of hinge legs were scissored to extract DNA to quantify the genotypes. The mRNA and protein levels of *henna* were then quantified in brains and fat bodies.”

6. Brain injections of dsRNA are notoriously difficult because of the impact of the injection-related injuries on behavior. The authors should include detailed protocols for how they achieved successful dsRNA injections, how did they choose which animals to include in their analysis, and what approaches were used to ensure precise targeting of specific brain regions. Including brain sections that show injection site would increase confidence in the included data.

Response #6: Thank you for your suggestion. The procedures for dsRNA delivery into locust brains were described in our previous study (Yang et al., PLoS Genet, 2014). The migratory locust possesses a highly sensitive and systemic RNAi response (Luo et al., Insect Mol Biol, 2013), capable of spreading expression silencing from the injection sites to distant cells in the brain.

To clarify this, we added the following sentences to the version:

“Fourth instar nymphs were positioned in a Kopf stereotaxic frame specifically adapted for locust surgery. Using Nevis scissors, a midline incision approximately 2 mm long was made at the midpoint between the two antennae, exposing the underlying brain. A total of 2 μ g of dsRNAs (2 μ g/ μ l) for *henna* TSS1, TSS2, and

GAF were injected into the brain using a glass micropipette tip mounted on a nanoliter injector (World Precision Instruments) under an anatomical lens. The dsRNAs of GFP were used as the negative controls. The injected locusts were returned to their normal rearing conditions and kept for 48 hours before their brains were harvested for RNA or protein extraction, or for behavior assays. We used randomization to assign individual locusts to the dsRNA groups to avoid selection bias and blinded the observer to the dsRNA groups during data collection and analysis to reduce subjective bias. The migratory locust possesses a highly sensitive and systemic RNAi response, capable of spreading expression silencing from the injection sites to distant cells in the brain (43).”

7. Imaging data shown in Fig. 4L are oversaturated and noninformative.

Response #7: Thanks for reminding us of this point. We substituted Fig. 4L with a new image. The new image was from a frozen brain section of a 4th instar nymph, stained with FITC for GAF and Alex-546 for PAH.

Accordingly, we added the following sentences to the version:

“A combined *in situ* analysis of the proteins PAH and GAF was conducted in the brains of 4th instar nymphs of gregarious locusts. Brains were dissected under an anatomical lens and fixed in 4% paraformaldehyde overnight. Brain sections (5 μ m thick), embedded in OCT (Sakura Tissue-Tek), were then washed twice in PBS for 5 minutes each. The sections were incubated in PBST (0.2% TritonX-100 in PBS) for 20 minutes, followed by two washes in PBS for 5 minutes each. The sections were then incubated in 5% bovine serum albumin in PBST at 25°C for 60 minutes, and then incubated with the polyclonal antibody GAF (1:50) at 37°C for 60 minutes, followed by three washes with PBS. The slides were then incubated with goat (Fab)-anti-rabbit FITC (1:500, Abcam, ab7050) at 37°C for 30 minutes. After being washed three times, the slides were incubated with the polyclonal antibody PAH (1:100) at 4°C overnight. After washing, the slides were incubated with goat -anti-rabbit Alexa Fluor 546 (1:1000, Life Technologies, A-11035) at 37°C for 30 minutes. The target signals were detected using an LSM 719 confocal fluorescence microscope (Zeiss).”

Figure 4M. Immunofluorescence of PAH and GAF proteins in the brains of 4th-instar nymphs of gregarious locusts. Blue, Hoechst; Green, GAF; Red, PAH. Scale bar, 20 μ m.

8. Because the orthologs of GAF and Grh are not currently annotated in the genome of *Locusta migratoria* genome, providing the specific LOC number of each identified gene is important.

Response #8: Thanks for your suggestion. We added the LOC number of GAF and Grh in the main text: “Intriguingly, multiple binding sites of GAGA factor (GAF, LOCMI11683) and Grainy head (Grh, LOCMI12694) were located within DHS1 sub-region II (Fig. 4E).” Actually, we verified the coding sequences of GAF and Grh by RT-PCR (reverse transcription-polymerase chain reaction) and Sanger sequencing to guarantee the accuracy of the involved genes.

Minor comments:

1. Phenotypic plasticity is not defined by the ability of a single genome to produce distinct phenotypes. In the case of locust, the two different phases should be defined

as "behavioral polyphenism", which is the capacity of the genome to encode two alternative behavioral states.

Response #9: Thanks for your suggestion, which helps us use words more accurately.

The following sentences were revised to the current forms:

“Our study reveals a novel epigenetic mechanism linking chromatin regulation with behavioral polyphenism in insects during environmental changes.”

“Phenotypic plasticity, the capacity of a single genome to produce distinct phenotypes under different environmental conditions, exhibits clear epigenetic regulatory characteristics (1).”

“Our study reveals open chromatin dynamics in regulating behavioral changes of locusts and provides insights into the epigenetic mechanisms underlying regulation of behavioral polyphenism in insects.”

“Mechanistically, the three GAF binding sites collaborate in the transcription initiation regulation of *henna* within four hours, demonstrating the capacity for rapid GAF-dependent chromatin regulation of behavioral polyphenism in short timescales.”

2. It is not clear what the sentence "Selection of *henna* promoter usage determines the playground of PAH enzymes" means (line 403).

Response #10: We revised the sentence in response to your comment.

“The selection of *henna* promoter usage determines the organs/tissues where PAH enzymes function.”

3. The relevance of PAH phosphorylation states in humans to the discussion of the data presented here is not obvious (lines 409-418).

Response #11: We removed the following sentence in the version:

“Human PAH with phosphorylated serine-16 is two to four times more active than the unphosphorylated enzyme.”

4. It is not clear why the authors claim that "Synergetic binding in multiple GAF

binding sites facilitates the rapid regulation of open chromatin" (lines 464-465). I though that the model is that once chromatin is open, synergy between enhancer elements can regulate expression. Unless I have missed it, the presented data do not show that GAF binding of DNA regulates chromatin openness.

Response #12: Thank you for your comments. Sorry for this unclear expression. We clarify that the findings in our manuscript are not related to any enhancers. The main findings of this manuscript are that the gregarious behavior of locusts is mediated by GAF-dependent regulation of chromatin openness on the brain-specific transcription initiation of the *henna* promoter through the synergistic collaboration of three GAF binding sites in DHS1. In the last two sub-sections of the Results section, we demonstrate that three GAF binding sites (GAF28, GAF60, and GAF166) in DHS1 serve two regulatory functions: activating promoter activity (GAF60) and maintaining chromatin openness (GAF28 and GAF166). Therefore, the three GAF binding sites, serving distinct regulatory functions, synergistically contribute to the increased TSS1 transcription observed in gregarious locusts compared to solitary locusts. In order to clarify this point, we provide a graphical summary to enhance the understanding of our work.

Referee #2:

In their work, Li, Jiang, and colleagues investigate the epigenetic basis for gene expression and behavior in the migratory locust *Locusta migratoria*. More specifically, they dissect pre-transcriptional regulation of the gene *henna* via GAGA factor (GAF) mediated changes in chromatin conformation at cis-regulatory sites. Through a series of highly detailed genomic, in vitro, in vivo, and behavioral experiments, their work demonstrates GAF binding to promoter regions of *henna* operates in a tissue specific manner to modulate gene expression associated with behavioral phase changes in locusts. Overall, this is an impressive body of work that integrates innovative and highly detailed methodology (especially by non-model system standards) and makes significant conceptual advances in determining the mechanistic basis by which chromatin conformation modulates gene expression and insect behavior, a research area of much needed attention. While I have a few suggestions on the presentation of the results, the findings are laid out in a highly logical and elegant fashion that makes understanding the results very accessible. Publication of this work will provide significant conceptual and methodological advances in the field of behavioral

plasticity, gene regulation, and perhaps in applied settings such as pest management. I have some major but primarily minor comments (presented chronologically) that I hope will improve the clarity of the results' presentation and interpretation.

Major comments:

1) A crucial part of many experiments focuses on the 4 hour exposure period of crowdedness or solitude to induce a behavior response. I think it would be helpful to explain or cite how this 4 hour period was determined because to a non-locust researcher this period of time may seem arbitrary.

Response #13: Thanks for this suggestion. Previous studies have demonstrated that the 4-hour exposure is sufficient to induce behavioral-related transcriptional changes of *Henna* involved in dopamine production (Ma et al., Proc Natl Acad Sci U S A, 2011; Yang et al., PLoS Genet, 2014). Therefore, we added the following sentences in the main text:

“Since 4 hours of isolation or crowding is sufficient to induce changes in the mRNA level of *henna* (6, 25), we set the 4-hour time point after isolation or crowding to quantify the pre-transcriptional expression change of *henna*.”

2) Lines 349-351/ Figure 6F: The authors state increasing GAF resulted in a gradual decrease in tri-nucleosome occupancy. However, this result is not quantified and the figure panel referenced (6F), to me, does not show any detectable decrease in tri-nucleosome occupancy. Lanes 1-3 do not look particularly dynamic or different and only lane 4 (highest GAF amount) is a qualitatively different bad pattern shown. If I'm interpreting this correctly, I would ask the authors to amend their writing to reflect this distinction, or lack thereof, between lanes as I'm not sure a dose-dependent relationship is shown in 6F. If I'm incorrectly interpreting this, please provide some sort of quantification to demonstrate tri-nucleosome occupancy is decreasing as GAF concentration increases. Furthermore, please specify the range GAF quantity used in the pink ramp-up part of the figure.

Response #14: Thanks for your valuable suggestions to improve our research quality. To precisely quantify the change in nucleosome occupancy, we conducted a DNA

fragment quantitative analysis using the Agilent 2100 Bioanalyzer system. Upon increasing the GAF/His amount (0, 0.1, 0.5, and 1 μg), the ratio of the digested DNA fragment (350 bp vs. 180 bp) decreased significantly. Thus, the present results indicated that the di-nucleosome occupancy decreased as the GAF amount increased. Therefore, we rewrote the sentence in the maintext:

“The gradual depletion of di-nucleosomes compared to mono-nucleosomes with the increases of GAF amount indicated that GAF reduced the density of positioned nucleosomes of DHS1 (Figs. 6F and Appendix Figure S9, $P_s < 0.0100$, Student's t-tests).”

Figure 6F: An agarose gel electrophoresis showing the gradual depletion of tri-nucleosomes at DHS1 as the GAF/His amount increases. The relative ratio of DNA amounts for di-nucleosomes versus mono-nucleosomes (left) at DHS1 ($n = 3$ for each group) was determined using the Agilent 2100 Bioanalyzer system.

Appendix Figure S9. Example diagrams of DNA fragment quantitative analysis assays using the Agilent 2100 Bioanalyzer system for different GAF amounts, including 0, 0.1, 0.5, 1 µg.

Minor comments:

A) Introduction

1) Lines 97-98: well said

Response #15: The sentence was revised as follows:

“However, the chromatin remodeler responsible for regulating transcription initiation, which in turn drives the development of different plastic phenotypes, remains elusive.”

B) Results

2) Line 117: "open chromatin"

Response #16: Corrected as suggested.

3) Line 122: switch order of enrichment and upstream.

Response #17: Corrected as suggested.

4) Lines 135-137: The authors state H3K36me3 deposition starts at DHS1 and spreads to coding sequence. However, examining Figure 1 shows a vanishingly small peak of H3K36me3 at DHS1 that looks little more than noise. Perhaps stating this mark occurs downstream of DHS1 is more accurate.

Response #18: Thank you for your suggestion. Corrected as suggested.

“Moreover, the deposition of H3K36me3 occurred downstream of DHS1 and spread into the coding region of *henna*.”

5) Line 137: "suggested" not "demonstrated"

Response #19: Corrected as suggested.

6) Lines 170: "Ho TSS3" is confusing, not sure what "Ho" is referring to

Response #20: Sorry for this. The word was corrected.

7) Line 283: "sites" typo

Response #21: Corrected as suggested.

8) Line 284: "precluded" perhaps too strong. "inhibited" is more appropriate

Response #22: Corrected as suggested.

C) Discussion

9) Line 408: please specify what percent identity defines a "large majority"

Response #23: Thank you for your comments. We added the percentage of the shared

amino acid sequence in the revision.

“The TSS1-driven *henna* isoform in brains, which has a unique serine-9 at the N-terminal tail, shares a large majority (96.5% of the total amino acids) of protein sequence with that driven by TSS2 in fat bodies.”

10) Lines 431-432: please cite relevant work for this claim'

Response #24: The reference was cited here in the revision.

Simola DF, Graham RJ, Brady CM, Enzmann BL, Desplan C, Ray A, Zwiebel LJ, Bonasio R, Reinberg D, Liebig J, Berger SL. Epigenetic (re)programming of caste-specific behavior in the ant *Camponotus floridanus*. *Science*. 2016 Jan 1;351(6268):aac6633.

11) Is there a reason why aggregation pheromone wasn't applied in this study? Given the behavioral work it isn't necessary, but could be a nice complement to the molecular and organismal results

Response #25: Thank you for your suggestion. Aggregation pheromones, which are recognized by odorant receptors expressed in olfactory receptor neurons housed under sensilla, serve their function in the peripheral nervous system (PNS). Specifically, 4VA has been identified as an aggregation pheromone of the migratory locust. OR35 is a specific odorant receptor for 4VA, and its knockout mutants lose attraction to 4VA. 4VA elicits responses specifically from basiconic sensilla, which are specialized sensory structures located on the antennae. The dopamine biosynthesis pathway in the brain, a critical pathway that modulates behavioral changes from the gregarious to the solitary phase, serves its function in the central nervous system (CNS). Exploring the connection between PNS and CNS in the context of olfactory inputs to behavioral neural regulation is very interesting. However, these processes are involved in complicated and multiple steps of sensory processing, neural integration, and behavioral output. We will address this interesting question in neuroscience in the future. For further details, please refer to our recent review paper (Guo and Kang, *Annu Rev Entomol*, 2024).

12) Line 460-463: This statement is claiming too much in my opinion by comparing the relative contribution of GAF to CBP in chromatin regulation, as this study did not

assess CBP in locusts. Please consider revising

Response #26: Based on your comments, the following sentences were removed in the revision:

“Moreover, caste-specific foraging and scouting behaviors are regulated epigenetically by CBP-mediated histone acetylation in ant brains (31). The inhibitor treatment of CBP prevents workers from reproducing and living longer following queen removal in the rock ant (34). Mechanistically, CBP functions as a histone acetyltransferase, directly promoting the remodeling of chromatin and the openness of chromatin (35).”

13) Line 484: typo "GAT"

Response #27: Corrected as suggested.

D) Methods

14) Lines 558-559: please specify what "narrow parameters" were used in MACS

Response #28: Thank you for your comments. The term "narrow parameters" should actually be "narrow peak parameters." This means that the "--broad" and "--broad-cutoff" options were not used in peak calling. We changed “using the narrow parameters” to “using the narrow peak parameters without the --broad and --broad-cutoff options” in the revision.

E) Figures

15) I'd highly recommend switching the order of panels in Figure 1F to that the DHS1 result is on top and DHS2 result is on bottom. This will align better with the brain vs fat body panels presented in Figure 1E

Response #29: Thank you for your suggestion. Figure 1F was reorganized as suggested.

16) Throughout the figures "a" "b" "bc" and "c" are noted on top on bar graphs. Please specify what these mean in the legend

Response #30: Thank you for your comments. The use of letters represents statistical differences between groups in multiple comparisons following an ANOVA. Same letter indicates that there is no significant statistical difference between the groups. Different letters indicates that there is a significant statistical difference between the

groups. Combined letters indicate groups that are statistically different from others, but are not significantly different from each other.

Therefore, we added the following sentences to the legend of the involved figures:

“Same letters above the bars indicate no significant difference between groups. Different letters indicate a significant difference between groups. Combined letters indicate groups are different from others but not from each other.”

Reservation: My background in insect behavioral studies is lacking, so I have refrained from providing feedback on the experimental behavior results.

Referee #3:

General summary

The undertaken study is tackling how short-term epigenetic regulation might account for the gregarious or solitary behavior of locusts upon response to environmental changes, which defines a challenge of interest for crossing scales from behavior and phenotypic plasticity down to molecular levels regulating an important regulator of dopamine involved in gregarious locusts, namely henna. In particular, the authors identify a putative enhancer that is differentially accessible upon such environmental change. They show that among the possible TFs involved, GAF is a bona fide candidate. GAF actually mediates chromatin accessibility of the corresponding enhancer and is able to drive the expression of henna.

While it is not surprising that chromatin accessibility is linked to the specific changes of chromatin associated with developmental cues and enhancers, the authors still characterize a complex system from behavior involving a central gene in the process down to molecular scales involving GAF. An ideal demonstration would be to demonstrate that without GAF, locusts are truly unable to modulate their collective behavior though the outcome of such an experiment can be obviously complex.

Major points:

1. Fig 1: It is not clear how many genes harbor several DHS in their bodies, and if Henna is in that sense particular. Furthermore, the brief analysis of histone marks is of correlative nature, and so the authors can hardly claim that: « These chromatin

characteristics demonstrated that DHS1 was the promoter region that initiated the predominant transcription of *henna* in the brains of gregarious locusts. » There is nothing like a demonstration in the finding that DHS1 is the key element to initiate predominant transcription. This is more a matter of transcriptional induction, not simply of chromatin accessibility.

Response #31: Thank you for your comments. We further clarify this question based on your comments. Please see the figure below that shows the percentage of genes with the number of peaks identified by MACS2 in the brain of the migratory locust. The average number of peaks is 2.62, with a 95% confidence interval of 1 to 11, which is lower than that in *henna*.

We agree with you that claiming the predominant transcription of *henna* in DHS1 is unfounded here. Indeed, the predominant transcription of *henna* in DHS1 was later confirmed by 5'-Cap RNA-seq and qRT-PCR assays, as shown in Figures 2B and 2C. Therefore, we removed the following sentence in the revision:

“These chromatin characteristics suggested that DHS1 was the promoter region that initiated the predominant transcription of *henna* in the brains of gregarious locusts.”

2. To assess the influence of DHS1 on transcription per se, the authors may probe nascent RNAs instead of mRNAs that also depends on RNA stability (Figure 1E). Given the position of DHS1 within an intron, this could be another mechanism of action that cannot be excluded.

Response #32: Thanks for your suggestion to improve our research. To assess the influence of DHS1 on transcription initiation per se, we probed the expression of nascent RNAs by quantifying the chromatin-associated RNAs, which are freshly synthesized RNAs and remain bound to transcription sites. The new results showed that, compared to wild-type controls, the nascent RNA expression in DHS1 was significantly reduced in the brains of the homozygous DHS1-dysfunctional mutants of gregarious locusts, while it didn't change significantly in DHS2. Given that the transcription starting from DHS1 was the main source of *henna* in the brain, we confirmed the influence of DHS1 on transcription initiation of nascent RNAs rather than on mRNA stability mediated by the intronic regulation of DHS1.

Therefore, we added this sentence in the revision:

“Moreover, to assess the influence of DHS1 on transcription initiation per se, we probed the expression of nascent RNAs by quantifying the chromatin-associated RNAs, which are freshly synthesized RNAs and remain bound to transcription sites. The results showed that, compared to wild-type controls, the nascent RNA expression in DHS1 was significantly reduced in the brains of the homozygous DHS1-dysfunctional mutants of gregarious locusts, while it didn't change significantly in DHS2 (**Fig. EV1B**, for TSS1, $P = 0.0286$, Mann Whitney test; for TSS2, $P = 0.4221$, Student's t-test).”

In the Methods section, we added the following description:

“Freshly dissected brains of homozygous DHS-mutated and gregarious locusts were frozen in liquid nitrogen and then homogenized by a Dounce grinder. Nuclei were collected by centrifugation and washed twice in the washing buffer (1mM EDTA, 0.1% Triton X-100, 1× protease inhibitor mix, 0.025mM a-amanitin, and SUPERase in PBS). After the supernatant was removed, nuclei were resuspended in glycerol buffer

(20mM pH8.0 Tris-HCl, 70mM NaCl, 0.5mM EDTA, 50% filter-sterilized glycerol, 0.84mM DTT, 1× protease inhibitor mix, 0.025mM a-amanitin, and SUPERase in RNase-free H₂O) and then lysed on ice for 2 min in nuclei lysis buffer (1% NP-40, 20mM pH7.5 HEPES, 0.2mM EDTA, 300mM NaCl, 1M filter-sterilized urea, 1mM DTT, 1× protease inhibition mix, 0.025mM α-amanitin, and SUPERase in RNase-free H₂O). After centrifugation and removal of the supernatant, the collected chromatins were resuspended in chromatin resuspension solution (1× protease inhibition mix, 0.025mM a-amanitin, and SUPERase in PBS). The RNA extraction and quantitative PCR analysis were the same as those previously mentioned.”

Fig. EV1B. The chromatin-associated RNA level of TSS1 and TSS2 in the brains of homozygous DHS1-dysfunctional mutants of gregarious locusts. Primers were designed specifically for the TSS. N = 4 for TSS1, and n = 5 for TSS2. WT, wildtype.

3. While the presence of GAF at TSS1-unlike TSS2- is of interest, it should be tested whether its binding is impaired in the mutant DHS1.

Response #33: Thanks for your suggestion. We accepted your suggestion to confirm confirm the impaired binding of GAF in the homozygous DHS1-dysfunctional mutants of gregarious locusts. Compared to the inaccessible chromatin region, the binding in the DHS1 mutants was significantly more impaired than that in the wild-type controls of 4th instar nymphs of gregarious locusts (Figure 4I). Moreover, we conducted an EMSA to confirm that its binding occurs directly through the three

GAF binding sites (GAF28, GAF60, and GAF166). When the GAF binding sites were mutated, the competitive binding from these mutated sites to the biotin-labeled sites was absent (Figure 4H).

In the version, we added these sentences:

“*In vitro*, GAF showed a direct interaction with DHS1 via the predicted GAF binding sites, as proven by the electrophoretic mobility shift assay (EMSA) (Fig. 4H).”

“Moreover, the chromatin immunoprecipitation PCR (ChIP-PCR) assays showed that the GAF binding in the homozygous DHS1-dysfunctional mutants was significantly more impaired than that in the wild-type controls (for WT, $P = 0.0018$; for DHS1-mutated, $P = 0.8726$, Student's t-test, **Figs. 4I and EV4**)”

Figure 4I. The fold enrichment of GAF binding at DHS1 *in vivo* in gregarious and DHS1-mutated nymphs proved by chromatin immunoprecipitation PCR (n = 5 for each group). Ctrl represents an inaccessible chromatin region.

Figure 4H. The direct interaction between GAF and its binding sites on DHS1 analyzed using EMSA. Bio/probes are biotin-labeled DHS1, while M_probes are DHS1 with mutated GAF binding sites. The red arrow indicates the target band.

4. Although other factors might be involved, the flow of arguments involving GAF binding to DHS1 is documented. However, ATAC-seq would have been more convincing as a way to re-scan the whole pattern of DHS in the henna gene locus, depending on presence absence of GAF in the cells.

Response #34: Thanks for your suggestion. In general, DNase-seq and ATAC-seq are two commonly employed methods for genome-wide mapping of open chromatin regions. Both methods use cleavage enzymes, including DNase-I in DNase-seq and Tn5 transposase in ATAC-seq, to identify and cleave DNA within accessible chromatin regions. ATAC-seq is less laborious and technologically challenging than DNase-seq. However, the dimeric Tn5 enzyme in ATAC-seq recognizes a wider region when interacting with DNA, thereby introducing sequence biases associated with Tn5 (Wolpe et al., NAR Genom Bioinform, 2023). Nevertheless, the use of DNase-seq in this study serves the same purpose as using ATAC-seq. Therefore, we think re-scanning open chromatin regions using ATAC-seq is not necessary within the

scope of this study. Furthermore, as shown in Figure 6D, the intensities of chromatin openness were significantly reduced in DHS1 after the knockdown of GAF expression. This is consistent with the observation that the knockdown of GAF expression significantly decreased TSS1 expression, as shown in Figure 4I. Both results confirmed that the absence of GAF expression in the brain is associated with decreased chromatin openness and transcription initiation at DHS1. We hope our answer will be satisfactory to you.

5. There is not direct demonstration that GAF is truly required to drive the change of behavior of locusts, though it is shown to be involved in DHS1-mediated expression of henna, which is required to such collective behavior (Fi. 3A). Re-assessing the gregarious behavior of locusts upon GAF depletion or after CRISPR of GAF sites from DHS1 would be needed to reach such a conclusion.

Response #35: Thanks for your suggestion. In Figures 4I, 4J and 4K, we verified the function relationships between GAF and TSS1 by the examination of TSS1 expression after the knockdown of GAF expression. The knockdown of GAF expression in gregarious locusts significantly decreased the TSS1 expression by at least 70% (**Fig. 4J**, $P < 0.0001$, Student's t-test). In particular, accompanied by the decrease in TSS1 expression, the locust gregarious behavior changed significantly, as reflected by reductions in Pgreg, distance moved and attraction index (**Figs. 4K and 4L**, for distance moved, $P = 0.0472$, and for attraction index, $P < 0.0001$, Wilcoxon rank sum exact test). As anticipated, the TSS2 expression was not affected in gregarious locusts subjected to GAF knockdown ($P > 0.0500$, Student's t-test). To exclude the possibility that GAF interacts to other genes in the dopamine synthesis pathway, we predicted GAF binding sites in the upstream regions of *pale*, *Ddc*, and *tan*, which are three other crucial genes in the dopamine synthesis pathway. Despite identifying a limited number of GAF binding sites (Appendix Figure S7), their location outside open chromatin regions inhibited their regulatory potential of GAF with these genes *in vivo*, thereby indicating an exclusive role of GAF specifically on TSS1.

Minor points:

1. I'm not sure of the terms « wild type compared to gregarious locusts », if gregarious is a behavior that is context dependent (but has the same genotype)?

Response #36: Thanks for your comments. Sorry for this confusion. Gregarious and solitary locusts have the same genotypes, but are developed under different rearing conditions that vary in population density. Wild-type gregarious locusts are locust nymphs reared at high population density without any genetic manipulation, including RNAi and CRISPR editing. dsRNA-treated gregarious locusts are locust nymphs subjected to RNAi treatments and maintained at high population density both before and after injection of dsRNAs.

2. Figures were not numbered.

Response #37: Thank you for your reminder. We have checked all the figures, and they are all numbered in this version.

3. In Fig 6b, a control is needed to test if GAF depletion is specifically altering DHS1 as opposed to another site or locus as every measure shows a decrease of H3.

Response #38: Thanks for your suggestion. In the revised Figure 6b, we added a experiment to show that GAF depletion did not cause a decrease in Histone H3 at DHS2 compared to control.

Figure 6B. The chromatin-nucleosome bound fractions of DHS1 sub-regions and DHS2 in G ctrl and RNAi GAF (n = 4 for each group).

References:

- Bailey, T. L., M. Boden, F. A. Buske, M. Frith, C. E. Grant, L. Clementi, J. Ren, W. W. Li, and W. S. Noble. 2009. 'MEME SUITE: tools for motif discovery and searching', *Nucleic Acids Res*, 37: W202-8.
- Castro-Mondragon, J. A., R. Riudavets-Puig, I. Rauluseviciute, R. B. Lemma, L. Turchi, R. Blanc-Mathieu, J. Lucas, P. Boddie, A. Khan, N. Manosalva Perez, O. Fornes, T. Y. Leung, A. Aguirre, F. Hammal, D. Schmelter, D. Baranasic, B. Ballester, A. Sandelin, B. Lenhard, K. Vandepoele, W. W. Wasserman, F. Parcy, and A. Mathelier. 2022. 'JASPAR 2022: the 9th release of the open-access database of transcription factor binding profiles', *Nucleic Acids Res*, 50: D165-D73.
- Guo, X., and L. Kang. 2024. 'Phenotypic Plasticity in Locusts: Trade-Off Between Migration and Reproduction', *Annu Rev Entomol*.
- Jiang, X., E. Dimitriou, V. Grabe, R. Sun, H. Chang, Y. Zhang, J. Gershenzon, J. Rybak, B. S. Hansson, and S. Sachse. 2024. 'Ring-shaped odor coding in the antennal lobe of migratory locusts', *Cell*, 187: 3973-91 e24.
- Luo, Y., X. Wang, X. Wang, D. Yu, B. Chen, and L. Kang. 2013. 'Differential responses of migratory locusts to systemic RNA interference via double-stranded RNA injection and feeding', *Insect Mol Biol*, 22: 574-83.

- Ma, Z., W. Guo, X. Guo, X. Wang, and L. Kang. 2011. 'Modulation of behavioral phase changes of the migratory locust by the catecholamine metabolic pathway', *Proc Natl Acad Sci U S A*, 108: 3882-7.
- Xiao, A., Z. Cheng, L. Kong, Z. Zhu, S. Lin, G. Gao, and B. Zhang. 2014. 'CasOT: a genome-wide Cas9/gRNA off-target searching tool', *Bioinformatics*, 30: 1180-82.
- Yang, M., Y. Wei, F. Jiang, Y. Wang, X. Guo, J. He, and L. Kang. 2014. 'MicroRNA-133 inhibits behavioral aggregation by controlling dopamine synthesis in locusts', *PLoS Genet*, 10: e1004206.
- Bailey, T. L., M. Boden, F. A. Buske, M. Frith, C. E. Grant, L. Clementi, J. Ren, W. W. Li, and W. S. Noble. 2009. 'MEME SUITE: tools for motif discovery and searching', *Nucleic Acids Res*, 37: W202-8.
- Castro-Mondragon, J. A., R. Riudavets-Puig, I. Rauluseviciute, R. B. Lemma, L. Turchi, R. Blanc-Mathieu, J. Lucas, P. Boddie, A. Khan, N. Manosalva Perez, O. Fornes, T. Y. Leung, A. Aguirre, F. Hammal, D. Schmelter, D. Baranasic, B. Ballester, A. Sandelin, B. Lenhard, K. Vandepoele, W. W. Wasserman, F. Parcy, and A. Mathelier. 2022. 'JASPAR 2022: the 9th release of the open-access database of transcription factor binding profiles', *Nucleic Acids Res*, 50: D165-D73.
- Guo, X., and L. Kang. 2024. 'Phenotypic Plasticity in Locusts: Trade-Off Between Migration and Reproduction', *Annu Rev Entomol*.
- Luo, Y., X. Wang, X. Wang, D. Yu, B. Chen, and L. Kang. 2013. 'Differential responses of migratory locusts to systemic RNA interference via double-stranded RNA injection and feeding', *Insect Mol Biol*, 22: 574-83.
- Ma, Z., W. Guo, X. Guo, X. Wang, and L. Kang. 2011. 'Modulation of behavioral phase changes of the migratory locust by the catecholamine metabolic pathway', *Proc Natl Acad Sci U S A*, 108: 3882-7.
- Wolpe, J. B., A. L. Martins, and M. J. Guertin. 2023. 'Correction of transposase sequence bias in ATAC-seq data with rule ensemble modeling', *NAR Genom Bioinform*, 5: lqad054.
- Xiao, A., Z. Cheng, L. Kong, Z. Zhu, S. Lin, G. Gao, and B. Zhang. 2014. 'CasOT: a genome-wide Cas9/gRNA off-target searching tool', *Bioinformatics*, 30: 1180-82.
- Yang, M., Y. Wei, F. Jiang, Y. Wang, X. Guo, J. He, and L. Kang. 2014. 'MicroRNA-133 inhibits behavioral aggregation by controlling dopamine synthesis in locusts', *PLoS Genet*, 10: e1004206.

Dear Dr. Kang,

Thank you for submitting your manuscript for consideration by the EMBO Journal. It has now been seen by three referees whose comments are enclosed. As you will see, all three referees express interest in your manuscript and are broadly in favour of publication, pending satisfactory minor revision.

Given the referees' positive recommendations, I would like to invite you to submit a revised version of the manuscript, addressing the comments of Reviewer #3. Furthermore, I would like to take care of all the issues raised by our editorial assistants (please see below).

We generally allow three months as standard revision time. As a matter of policy, competing manuscripts published during this period will not negatively impact on our assessment of the conceptual advance presented by your study. However, we request that you contact the editor as soon as possible upon publication of any related work, to discuss how to proceed.

Thank you for the opportunity to consider your work for publication. I look forward to your revision.

Yours sincerely,

Yehu Moran
Academic Editor
The EMBO Journal

General Instructions for preparing your revised manuscript:

We realize that it is difficult to revise to a specific deadline. In the interest of protecting the conceptual advance provided by the

work, we recommend a revision within 3 months (12th May 2025). Please discuss the revision progress ahead of this time with the editor if you require more time to complete the revisions. Use the link below to submit your revision:

Specific issues raised by editorial assistants which require special attention:

MANUSCRIPT FORMAT: You should upload a .docx file with no figures and no track changes.

FUNDING INFO: The Funding section should be included in Acknowledgements section.

REFERENCE FORMAT: Should be alphabetical and have 10 authors listed + et al. instead of numbered with 1 author + et al.

Conflict of Interest: title needs to be renamed as "DISCLOSURE AND COMPETING INTERESTS STATEMENT"

AC/CRedit: section needs to be removed from the text and inserted only via the system.

FIGURE CALLOUTS: missing callout for Appendix Table S1 and S2.

DATASET EV LEGENDS: n/a

APPENDIX 1 FILE WITH TABLE OF CONTENTS: Appendix file needs to be in PDF format; page numbers of the listed items missing in the Table of contents; references need to be corrected to be alphabetical and have 10 authors listed + et al. instead of numbered with 1 author + et al.; "Materials and Methods" need to be removed from Appendix PDF and included in main manuscript file with the section name "Methods".

SYNOPSIS IMAGE: included, but a bit too small (539x473 px) - should be 550 px wide.

DATA CHECK: PASS: Please note that the specific URL for (10.57760/sciencedb.18341) dataset is not provided in the data availability statement. This should be fixed.

- Figure Legends (main + EV): 1. Please note that the exact p-values are not provided in the legends of figures 1C-F; 3B-D; 5C-E; 6B, C, D, F, H, I, J; EV1 B, EV3 B, EV4. This should be corrected.

- Please indicate what */ **/ ***/ **** represents; if this represents p value(s), please indicate the statistical test used and where appropriate, specify the exact p value in the legend(s) of figure(s) 4I, J, L

- Please note that in figures EV3 B there is a mismatch between the annotated p values in the figure legend and the annotated p values in the figure file that should be corrected.

- Please note that the box plots need to be defined in terms of minima, maxima, centre, bounds of box and whiskers, and percentile in the legends of figures 3B, 4L.

- Please note that information related to n is missing in the legends of figures 3, 4L.

- Section order should be corrected: Title page - Abstract & Keywords - Introduction - Results - Discussion - Methods - Data Availability - Acknowledgements - Disclosure and Competing Interests Statement - References - Figure Legends - Table(s) - Expanded View Figure Legends.

- Subject categories missing. Please provide.

Referee #1:

The revised manuscript entitled "GAF-dependent Chromatin Plasticity Determines Promoter Usage to Mediate Locust Gregarious Behavior" by Li et al. describes the role of GAF-dependent regulation of henna transcription in locust behavioral polyphenism via an epigenetic mechanism. Understanding the genetic and epigenetic mechanisms that regulate behavior of the locust is important for basic and translational reasons. Furthermore, because the locust is not a genetically tractable model system, developing approaches for manipulating its genome provides an important technical step forward. The revised manuscript is significantly improved, and the authors did a good job in terms of addressing majority of previously identified weaknesses. In particular, addressing the missing methodological details improved my confidence in the presented data.

Referee #2:

I thank the authors for addressing my comments and have no further suggestions on the manuscript. It is an impressive body of work in my opinion.

Referee #3:

The authors have addressed most of the comments raised. Notably, the linkage between DS1 and gregarious locust is documented and the link between GAF recruitment and DHS1, the activity of TSS1 as well as of Henna is now clarified.

Comment: I still find that the paper would have been strengthened if they could further tackle how GAF binding to DS1 impede on the gregarious phenotype, i.e. using GAF mutants. An experiment showing that tethering of GAF to TSS1 is sufficient to restore the gregarious phenotype would be a demonstration, and exclude other possibilities. Without such an experiment, the authors cannot exclude additional modes of action involving GAF as a master regulator. Actually, a high proportion of genes harbor multiple GAF sites. So, the authors should at least discuss that they cannot rule out the possibility that GAF affects a phenotype solely based on DS1 and on Henna.

Dear Professor Moran,

On behalf of my co-authors, I would like to express our sincere appreciation to you and the reviewers for your positive assessment of our paper. We have revised our paper based on the comments provided by Reviewer #3 and the editorial assistants.

Looking forward to hearing from you.

Best regards,

Le Kang, Ph.D.

CAS Distinguished Professor

Institute of Zoology

Chinese Academy of Sciences (CAS)

Beijing 100101, China

Tel: 86-10-6480-7219

Fax: 86-10-6480-7099

E-mail: lkang@ioz.ac.cn

Response to the Reviewers

Explanation: All editorial correspondence from *EMBO J*, including the reviewers' comments, is verbatim in black. Our responses are inserted directly into this text in blue. All changes have been implemented in the final version of the manuscript.

Message: Dear Dr. Kang,

Thank you for submitting your manuscript for consideration by the EMBO Journal. It has now been seen by three referees whose comments are enclosed. As you will see, all three referees express interest in your manuscript and are broadly in favour of publication, pending satisfactory minor revision.

Given the referees' positive recommendations, I would like to invite you to submit a revised version of the manuscript, addressing the comments of Reviewer #3. Furthermore, I would like to take care of all the issues raised by our editorial

assistants (please see below).

We generally allow three months as standard revision time. As a matter of policy, competing manuscripts published during this period will not negatively impact on our assessment of the conceptual advance presented by your study. However, we request that you contact the editor as soon as possible upon publication of any related work, to discuss how to proceed.

Thank you for the opportunity to consider your work for publication. I look forward to your revision.

Yours sincerely,

Yehu Moran
Academic Editor
The EMBO Journal

General Instructions for preparing your revised manuscript:

We realize that it is difficult to revise to a specific deadline. In the interest of protecting the conceptual advance provided by the work, we recommend a revision within 3 months (12th May 2025). Please discuss the revision progress ahead of this time with the editor if you require more time to complete the revisions. Use the link below to submit your revision:

Specific issues raised by editorial assistants which require special attention:

MANUSCRIPT FORMAT: You should upload a .docx file with no figures and no track changes.

FUNDING INFO: The Funding section should be included in Acknowledgements section.

Responses: The funding information has been moved to the Acknowledgements section.

REFERENCE FORMAT: Should be alphabetical and have 10 authors listed + et al. instead of numbered with 1 author + et al.

Responses: The reference format has been adjusted based on this comment.

Conflict of Interest: title needs to be renamed as "DISCLOSURE AND COMPETING INTERESTS STATEMENT"

Responses: Correct as suggested.

AC/CRedit: section needs to be removed from the text and inserted only via the system.

Responses: Correct as suggested.

FIGURE CALLOUTS: missing callout for Appendix Table S1 and S2.

Responses: Correct as suggested.

DATASET EV LEGENDS: n/a

Responses: We do not include any datasets in the Expanded View.

APPENDIX 1 FILE WITH TABLE OF CONTENTS:

Appendix file needs to be in PDF format; page numbers of the listed items missing in the Table of contents;

Responses: The page numbers of the listed items are provided on Page 2 of the Appendix file.

references need to be corrected to be alphabetical and have 10 authors listed + et al. instead of numbered with 1 author + et al.;

Responses: The reference format has been adjusted based on this comment.

"Materials and Methods" need to be removed from Appendix PDF and included in main manuscript file with the section name "Methods".

Responses: Correct as suggested.

SYNOPSIS IMAGE: included, but a bit too small (539x473 px) - should be 550 px wide.

Responses: The image quality of the synopsis image was improved to fit this requirement.

DATA CHECK: PASS: Please note that the specific URL for (10.57760/sciencedb.18341) dataset is not provided in the data availability statement. This should be fixed.

Responses: The data download URL is provided in the Data Availability statement.

- Figure Legends (main + EV): 1. Please note that the exact p-values are not provided in the legends of figures 1C-F; 3B-D; 5C-E; 6B, C, D, F, H, I, J; EV1 B, EV3 B, EV4. This should be corrected.

Responses: Correct as suggested.

- Please indicate what */ **/ ***/ **** represents; if this represents p value(s), please indicate the statistical test used and where appropriate, specify the exact p value in the legend(s) of figure(s) 4I, J, L

Responses: Correct as suggested.

- Please note that in figures EV3 B there is a mismatch between the annotated p values in the figure legend and the annotated p values in the figure file that should be corrected.

Responses: Correct as suggested.

- Please note that the box plots need to be defined in terms of minima, maxima, centre, bounds of box and whiskers, and percentile in the legends of figures 3B, 4L.

Responses: Correct as suggested.

- Please note that information related to n is missing in the legends of figures 3, 4L.

Responses: Correct as suggested.

- Section order should be corrected: Title page - Abstract & Keywords - Introduction - Results - Discussion - Methods - Data Availability - Acknowledgements - Disclosure and Competing Interests Statement - References - Figure Legends - Table(s) - Expanded View Figure Legends.

Responses: We checked the section order to ensure it is consistent with this style.

- Subject categories missing. Please provide.

Responses: It is provided in the current submission.

Referee #1:

The revised manuscript entitled "GAF-dependent Chromatin Plasticity Determines Promoter Usage to Mediate Locust Gregarious Behavior" by Li et al. describes the role of GAF-dependent regulation of *henna* transcription in locust behavioral polyphenism via an epigenetic mechanism. Understanding the genetic and epigenetic mechanisms that regulate behavior of the locust is important for basic and translational reasons. Furthermore, because the locust is not a genetically tractable model system, developing approaches for manipulating its genome provides an important technical step forward. The revised manuscript is significantly improved, and the authors did a good job in terms of addressing majority of previously identified weaknesses. In particular, addressing the missing methodological details improved my confidence in the presented data.

Referee #2:

I thank the authors for addressing my comments and have no further suggestions on the manuscript. It is an impressive body of work in my opinion.

Referee #3:

The authors have addressed most of the comments raised. Notably, the linkage between DS1 and gregarious locust is documented and the link between GAF recruitment and DHS1, the activity of TSS1 as well as of *Henna* is now clarified.

Comment: I still find that the paper would have been strengthened if they could further tackle how GAF binding to DS1 impede on the gregarious phenotype, i.e. using GAF mutants. An experiment showing that tethering of GAF to TSS1 is sufficient to restore the gregarious phenotype would be a demonstration, and exclude other possibilities. Without such an experiment, the authors cannot exclude additional modes of action involving GAF as a master regulator. Actually, a high proportion of genes harbor multiple GAF sites. So, the authors should at least discuss that they cannot rule out the possibility that GAF affects a phenotype solely based on DS1 and on *Henna*.

Responses: We added the following sentence in the Discussion section:

“Due to the presence of multiple predicted GAF binding sites in the locust genome, we cannot rule out the possibility that GAF modulates the behavioral plasticity of locusts, not solely through DHS1 and *henna*.”

Dear Dr. Kang,

I am pleased to inform you that your manuscript has been accepted for publication in the EMBO Journal.

Yours sincerely,

Yehu Moran
Academic Editor
The EMBO Journal
